# A comprehensive assessment of current methods for measuring metacognition

Dobromir Rahnev [1,2] ✉

One of the most important aspects of research on metacognition is the measurement of metacognitive ability. However, the properties of existing measures of metacognition have been mostly assumed rather than empirically established. Here I perform a comprehensive empirical assessment of 17 measures of metacognition. First, I develop a method of determining the validity and precision of a measure of metacognition and find that all 17 measures are valid and most show similar levels of precision. Second, I examine how measures of metacognition depend on task performance, response bias, and metacognitive bias, finding only weak dependences on response and metacognitive bias but many strong dependencies on task performance. Third, I find that all measures have very high split-half reliabilities, but most have poor test-retest reliabilities. This comprehensive assessment paints a complex picture: no measure of metacognition is perfect and different measures may be preferable in different experimental contexts.

Metacognition is classically defined as knowing about knowing[1]. Within this broad construct, the term "metacognitive ability" refers more narrowly to the capacity to evaluate one's decisions by distinguishing between correct and incorrect answers[2,3]. High metacognitive ability allows us to have high confidence when we are correct but low confidence when we are wrong. Conversely, low metacognitive ability impairs the capacity of confidence ratings to distinguish between instances when we are correct or wrong. Metacognitive ability is thus a critical capacity in human beings linked to our ability to learn[4], make good decisions[5], interact with others[6], and know ourselves[7]. As such, it is critical that we have the tools to precisely measure metacognitive ability in human participants.

Metacognitive ability is typically assumed to be a somewhat stable trait with meaningful variability across people[2,8,9]. Consequently, metacognitive ability has been correlated with other stable individual differences, such as brain structure[10–13]. While metacognitive ability is often assumed to be domain-general and rely on shared neural substrates, this question remains hotly debated[14–17]. The construct of metacognitive ability is also thought to be different from other constructs such as task skill or bias, so it is often desirable to find metrics of metacognitive ability unrelated to these other constructs[18].

Below, I first examine the properties that one may desire in a measure of metacognition and then review the known properties of existing measures of metacognitive ability. This brief overview demonstrates that there is little we firmly know about the properties of existing measures of metacognition. The rest of the paper aims to fill this gap by providing a comprehensive test of the critical properties of many common measures of metacognition.

Before one can evaluate a given measure of metacognition, it is first necessary to determine what properties are important or desirable. Since there is no existing list of desirable properties, I start by creating one here (Supplementary Table 1) and discuss each property below.

The most important property of any measure is that it is valid: namely, it should measure what it purports to measure[19]. Existing measures of metacognitive ability assess the degree to which confidence is associated with objective reality, thus making them face valid. Still, we lack a formal way of verifying the validity of existing measures. A related property is precision. I use the term "precision" following its definitions in the literature as "the ability to repeatedly measure a variable with a constant true score and obtain similar results"[20], "the margin of error" in a measurement[21], or "the spread of values that would be expected across multiple measurement attempts"[22]. Note that precision here does not refer to whether a

[1]School of Psychology, Georgia Institute of Technology, Atlanta, GA, USA. [2]Computational Cognition Center of Excellence, Georgia Institute of Technology, Atlanta, GA, USA. ✉e-mail: rahnev@psych.gatech.edu

measure is only affected by the construct of interest. Precision has been largely ignored in the context of measures of metacognition and we currently lack methods to measure it. Here I develop a simple and intuitive method for assessing both validity and precision of metacognition measures. The method demonstrates that all existing measures of metacognition are valid but show some variations in precision.

Another critical property of measures of metacognition—one that is perhaps the most widely appreciated—is that such measures should be independent of various nuisance variables. Here a "nuisance variable" is any property of people's behavior that is not directly related to their metacognitive ability.

The nuisance variable that has received the most attention is task performance. It is often desirable that a measure of metacognition should not be affected by whether people happened to be performing an easy or a difficult task[3,18]. For example, in visual perception tasks with confidence, there is little reason to believe that the underlying metacognitive ability should be affected by stimulus contrast. Thus, one may want to measure the same metacognitive ability regardless of contrast level. Note that there are subtleties here. If difficulty is manipulated by introducing cognitive load or other task demands that may tax the metacognitive system, then one would not necessarily expect metacognitive ability to remain the same anymore (though whether metacognitive ability is affected by working memory load remains a topic of debate[23–25]). Therefore, the logic here applies more readily to stimulus than task manipulations. That said, even if one does not agree that metacognitive ability should be independent from task performance, examining how each measure depends on task performance is still informative, especially if there are meaningful differences between measures. Task performance can be computed as $d'$, which is a measure of sensitivity derived from signal detection theory (SDT).

A second nuisance variable is response bias, that is, the tendency to select one response category more than another[18]. For two-choice tasks, this variable can be quantified as the decision criterion, $c$, derived from SDT. Response bias is under strategic control in that participants can freely choose to select one stimulus category more often than others. In fact, they consistently do so in response to experimental manipulations such as expectation or reward[26]. As such, measures of metacognitive ability should ideally remain independent of response bias.

The final nuisance variable is metacognitive bias, that is, the tendency of people to be biased towards the lower or upper ranges of the confidence scale[27,28]. This variable can be quantified simply as the average confidence across all trials. As with response bias, metacognitive bias is under strategic control in that participants can freely choose to use lower or higher confidence. As such, measures of metacognitive ability should ideally remain independent of metacognitive bias because we do not want to measure different ability if people purposefully choose to use predominantly low or high confidence ratings[3]. The logic here is similar to the logic in SDT, where the measure of performance ($d'$) is designed to be mathematically independent from the measure of response bias ($c$)[29]. In the case of SDT, we interpret high $d'$ values as showing high ability to perform the task even if the participant exhibits an extreme bias and, consequently, low percent of correct responses. Similarly, this paper, following the standard in the field[18], adopts the perspective that measures of metacognitive ability should be independent of metacognitive bias.

Task performance, response bias, and metacognitive bias are arguably the primary nuisance variables that a measure of metacognitive ability should be independent of (Supplementary Table 2). They are also variables that can be measured in any design that also allows the measurement of metacognitive ability. It is possible to add more variables to this list (e.g., reaction time[30]) but the current paper only examines these three variables.

The final critical property of measures of metacognition is that they should be reliable. This property is critical for studies of individual differences. This paper examines both split-half and test–retest reliability.

Having reviewed the desirable properties of measures of metacognition, let us now turn our attention to the existing measures of metacognitive ability. One popular measure is the area under the Type 2 ROC function[31], also known as $AUC2$. Other popular measures are the Goodman–Kruskall Gamma coefficient (or just $Gamma$), which is essentially a rank correlation between trial-by-trial confidence and accuracy[32] and the Pearson correlation between trial-by-trial confidence and accuracy (known as $Phi$[33]). Another simple but less frequently used measure is the difference between average confidence on correct trials and the average confidence on error trials (which I call $\Delta Conf$).

While all four of these traditional measures are intuitively appealing, they are all thought to be influenced by the primary task performance[18]. To address this issue, Maniscalco and Lau[34] developed a new approach to measuring metacognitive ability where one can estimate the sensitivity, $meta$-$d'$, exhibited by the confidence ratings. Because $meta$-$d'$ is expressed in the units of $d'$, Maniscalco and Lau then reasoned that $meta$-$d'$ can be normalized by the observed $d'$ to obtain either a ratio measure ($M$-$Ratio$, equal to $meta$-$d'/d'$) or a difference measure ($M$-$Diff$, equal to $meta$-$d'-d'$). These measures are often assumed to be independent of task performance[18].

The normalization introduced by Maniscalco and Lau[34] has only been applied to the measure $meta$-$d'$ (resulting in the measures $M$-$Ratio$ and $M$-$Diff$), but there is no theoretical reason why a conceptually similar correction cannot be applied to the traditional measures above. Consequently, here I develop eight new measures where one of the traditional measures of metacognitive ability is turned into either a ratio ($AUC2$-$Ratio$, $Gamma$-$Ratio$, $Phi$-$Ratio$, and $\Delta Conf$-$Ratio$) or a difference ($AUC2$-$Diff$, $Gamma$-$Diff$, $Phi$-$Diff$, and $\Delta Conf$-$Diff$) measure. The logic is that a given measure (e.g., $AUC2$) is computed once using the observed data (obtaining, e.g., $AUC2_{observed}$) and a second time using the predictions of SDT given the observed sensitivity and decision criterion (obtaining, e.g., $AUC2_{expected}$). One can then take either the ratio or the difference between the observed and the SDT-predicted quantities.

Finally, one important limitation of all measures above is that they are not derived from a process model of metacognition. In other words, none of these measures are based on an explicit model of how confidence judgments may be corrupted. Recently, Shekhar and Rahnev[27] developed a process model of metacognition—the lognormal meta noise model—that is based on SDT assumptions but with the addition of lognormally distributed metacognitive noise. This metacognitive noise corrupts the confidence ratings but not the initial decision and, in the model, takes the form of confidence criteria that are sampled from a lognormal distribution rather than having constant values. The metacognitive noise parameter ($\sigma_{meta}$, referred here as $meta$-$noise$) can then be used as a measure of metacognitive ability. A similar approach was taken by Boundy-Singer et al.[35] who developed another process model of metacognition, CASANDRE, based on the notion that people are uncertain about the uncertainty in their internal representations. The second-order uncertainty parameter ($meta$-$uncertainty$) thus represents another possible measure of metacognitive ability.

This paper examines the properties of all 17 measures of metacognition introduced above (for a summary, see Table 1). Before then, however, I briefly review the previous literature on the properties of these measures.

Given the importance of using measures with good psychometric properties, it is perhaps surprising that the published literature contains very little empirical investigation into the properties of the different measures of metacognition. For example, no paper to date has examined the precision of any existing measure. Several papers have relied exclusively on simulations to investigate some of the properties of measures of metacognition[36,37]. Such investigations are important but cannot substitute empirical studies because it is a priori unknown how well the process models used to simulate data reflect empirical reality. Evans and Azzopardi[38] empirically showed that a specific measure of metacognition, Kunimoto's $a'$[39], exhibits a strong

**Table 1 | Measures of metacognition examined in the current paper**

| Measure | Calculation | Based on a process model |
|---|---|---|
| meta-d' | d' value that provides best fit to Type 2 ROC | No |
| AUC2 | Area under the Type 2 ROC curve | No |
| Gamma | Rank correlation between confidence and accuracy | No |
| Phi | Pearson correlation between confidence and accuracy | No |
| ΔConf | Difference between average confidence for correct and error trials | No |
| M-Ratio | meta-d' divided by d' | No |
| AUC2-Ratio | AUC2 divided by expected AUC2 under SDT assumptions | No |
| Gamma-Ratio | Gamma divided by expected Gamma under SDT assumptions | No |
| Phi-Ratio | Phi divided by expected Phi under SDT assumptions | No |
| ΔConf-Ratio | ΔConf divided by expected ΔConf under SDT assumptions | No |
| M-Diff | meta-d' minus d' | No |
| AUC2-Diff | AUC2 minus expected AUC2 under SDT assumptions | No |
| Gamma-Diff | Gamma minus expected Gamma under SDT assumptions | No |
| Phi-Diff | Phi minus expected Phi under SDT assumptions | No |
| ΔConf-Diff | ΔConf minus expected ΔConf under SDT assumptions | No |
| meta-noise | Metacognitive noise computed using the lognormal meta noise model | Yes |
| meta-uncertainty | Metacognitive uncertainty computed using the CASANDRE model | Yes |

dependence on response bias. Because Kunimoto's $a'$ is built on wrong distributional assumptions[40], it is not investigated here. Finally, several older papers investigated the theoretical properties of several measures independent of any simulations or empirical data[32], but this approach cannot be used to establish the *empirical* properties of the measures under consideration.

Only recently, Shekhar and Rahnev[27] examined the dependence on both task performance and metacognitive bias for five measures: *meta-d'*, *M-Ratio*, *AUC2*, *Phi*, and *meta-noise*. They found that *meta-d'*, *AUC2*, and *Phi* strongly depend on task performance, but *M-Ratio* and *meta-noise* do not. On the other hand, *meta-d'*, *M-Ratio*, *AUC2*, and *Phi* have a complex dependence on metacognitive bias, while only *meta-noise* appeared independent of it. Guggenmos[41] examined both the split-half reliability and the across-participant correlation between $d'$ and several measures of metacognition (*meta-d'*, *M-Ratio*, *M-Diff*, and *AUC2*) finding surprisingly low reliability and significant correlations with d' for all measures. Relatedly, Kopcanova et al.[14] examined the test-retest reliability of *M-Ratio* and also found low-reliability values. Another paper developed a new technique to examine dependence on metacognitive bias and found that *meta-d'* and *M-Ratio* are not independent of metacognitive bias[28]. Finally, Boundy-Singer et al.[35] showed that *meta-uncertainty* appears to have high test–retest reliability, and only a weak dependence on task performance and metacognitive bias.

As this brief overview demonstrates, most previous investigations only focused on a few measures of metacognition, only examined a few of the critical properties of interest, and often did not make use of empirical data. Here, I empirically examine each of the critical properties for all 17 measures of metacognition introduced above. To do so, I make use of six large datasets[27,42–46] (Table 2) all made available on the Confidence Database[47]. All datasets involve 2-choice tasks because most measures of metacognition only apply to 2-choice tasks.

Overall, I find that no current measure of metacognitive ability is "perfect" in the sense of possessing all desirable properties. Nevertheless, they are not equivalent either with many important differences between measures emerging. Based on these results, I make recommendations for the use of different measures of metacognition based on the specific analysis goals.

## Results

Here I assess the properties of 17 measures of metacognition. Specifically, I focus on each measure's (1) validity and precision, (2) dependence

on nuisance variables, and (3) reliability. To examine each of these properties, I use six existing datasets (Table 2) from the Confidence Database. For each property, I analyze the data from between one and three of the six datasets. In addition, I compute precision and reliabilities using 50, 100, 200, or 400 trials at a time to clarify how these measures behave for different amounts of underlying data.

### Validity and precision

Perhaps the most important requirement for any measure is that it is both valid and precise[19–22,48]. In other words, a measure should reflect the quantity it purports to measure, and it should do so with a high level of quantitative accuracy. However, despite the importance of both criteria, there has been no formal method to assess the validity or precision of measures of metacognition.

Here I develop a simple method for assessing both properties. The method selects a small proportion of trials and decreases confidence by 1 point for each correct trial and increases confidence by 1 point for each incorrect trial. This manipulation artificially decreases the informativeness of confidence ratings. A valid measure of metacognition should therefore show a drop when applied to these altered data. The size of the drop relative to the normal fluctuations of the measure quantifies the precision of the measure (i.e., if the drop is large relative to background fluctuations, this indicates that the measure has a high level of precision).

To quantify the precision of existing measures of metacognition, one would ideally use a dataset with very large number of trials coming from a single experimental condition because mixing conditions can strongly impact metacognitive scores[49]. Consequently, I selected the two datasets from the Confidence Database with the largest number of trials per participant that also had a single experimental condition: Haddara (3000 trials per participant) and Maniscalco (1000 trials per participant). In each case, I examined the results of altering 2, 4, and 6% of all trials and computed metacognitive scores using bins of 50, 100, 200, and 400 trials.

The results showed that all 17 measures are valid in that metacognitive scores decreased when confidence ratings were artificially corrupted (Fig. 1). The decrease in each measure was roughly a linear function of the percent of trials corrupted. For example, in the Haddara dataset, the values of *meta-d'* decreased from an average of 1.14 without any corruption to averages of 0.98, 0.84, and 0.72 when 2%, 4%, and 6% of trials were corrupted, respectively (for an average drop of about 0.14 for every 2% of trials corrupted). However, this drop is

**Table 2 | Datasets used in the current paper**

| Dataset | Haddara | Locke | Maniscalco | Rouault1 | Rouault2 | Shekhar |
|---|---|---|---|---|---|---|
| # participants analyzed | 70 | 10 | 22 | 466 | 484 | 20 |
| # excluded participants | 5 | 0 | 8 | 32 | 13 | 0 |
| % excluded participants | 7% | 0% | 27% | 6% | 3% | 0% |
| # trials/participant | 3000 | 4900 | 1000 | 210 | 210 | 2800 |
| # total trials in experiment | 210,000 | 49,000 | 22,000 | 97,860 | 101,640 | 56,000 |
| # difficulty levels | 1 | 1 | 1 | 70 | staircase | 3 |
| Criterion manipulated | — | ✓ | — | — | — | — |
| Original confidence scale | 4-point | 2-point | 4-point | 11-point | 6-point | Continuous |
| Analyses of each dataset | | | | | | |
| Precision | ✓ | — | ✓ | — | — | — |
| Dependence on task performance | — | — | — | ✓ | ✓ | ✓ |
| Dependence on metacognitive bias | ✓ | — | ✓ | — | — | ✓ |
| Dependence on response bias | — | ✓ | — | — | — | — |
| Split-half reliability | ✓ | — | ✓ | — | — | ✓ |
| Test–retest reliability | ✓ | — | — | — | — | — |
| Inter-measure correlations | ✓ | — | ✓ | — | — | ✓ |

The table lists details of each dataset and indicates which analyses in the present paper each dataset was used for.

difficult to compare between measures because different measures are on different scales (e.g., *meta-d′* normally takes values between 0 and ∞, whereas *AUC2* normally takes values between 0.5 and 1). Therefore, to obtain values that are easy to interpret and compare, one can normalize the average drop after corruption by the standard deviation (SD) of the observed values across different subsets of trials in the absence of any corruption. Because the SD value is larger for smaller bin sizes—reflecting the larger noisiness of each measure when few trials are used—the results show that larger bin sizes lead to greater precision of the measures (Fig. 1a). Indeed, across the 17 measures, corrupting 2% of the trials led to an average decrease of 0.35, 0.50, 0.70, and 1.04 SDs in the measured metacognitive ability value for bins of 50, 100, 200, and 400 trials, respectively.

This technique allows us to compare the precision of different measures. To simplify the comparison, I averaged the decreases across the four different bin sizes and the three levels of corruption (2, 4, and 6%; Fig. 1b,c). These analyses revealed that the precision scores were overall higher in the Haddara compared to the Maniscalco datasets. This difference is likely due to differences in variables such as sensitivity and metacognitive bias that are likely to vary across datasets. Therefore, the technique introduced here is useful for comparing between different measures but is unlikely to be useful if one wants to compare values across different datasets.

More importantly, most measures of metacognition showed comparable levels of precision (Fig. 1b,c). The one exception was the measure *meta-uncertainty*, which had substantially lower average precision score in both the Haddara (*meta-uncertainty*: 0.37; average of other measures: 0.67; ratio = 0.56) and the Maniscalco datasets (*meta-uncertainty*: 0.30; average of other measures: 0.53; ratio = 0.58). Indeed, pairwise comparisons showed that, without multiple comparison correction, the precision for *meta-uncertainty* was significantly lower than every one of the other 16 measures in both datasets ($p < 0.05$ for all 32 comparisons). In the Haddara dataset, 15 of the 16 comparisons remained significant even after applying a very conservative Bonferroni correction for the existence of $\frac{17*16}{2} = 136$ pairwise comparisons; in the smaller Maniscalco dataset, no comparison remained significant after this conservative correction. This difference between *meta-uncertainty* and the remaining measures may stem from the noisiness of the process of estimating *meta-uncertainty* in the presence of relatively few trials. In fact, the original authors who introduced *meta-uncertainty* already warned

about the dangers of trying to compute this variable using low trial numbers[35].

The differences between the remaining measures were much smaller and, in some cases, inconsistent across the two datasets. The differences between all other measures of pairs were never significant (at $p < 0.05$ uncorrected) in both the Haddara and Maniscalco datasets. Nevertheless, there appear to be some small but consistent difference between measures, such that *meta-d′*, *Gamma*, *Phi*, *Gamma-Diff*, *Phi-Diff*, and *meta-noise* show above-average precision, whereas AUC2, *ΔConf*, and *ΔConf-Diff* show below-average precision (Fig. 1d). Overall, these analyses suggest that all measures of metacognition investigated here are valid, and that most have comparable level of precision except for *meta-uncertainty*, which appears to be noisier than the remaining measures. Whether the differences between the remaining measures are meaningful remains to be demonstrated.

## Dependence on nuisance variables
Beyond validity and precision, another important feature for good measures of metacognition is that they should not be influenced by nuisance variables. Here I examine three nuisance variables—task performance, metacognitive bias, and response bias—and test how much each of these variables affects each of the 17 measures of metacognition.

## Dependence on task performance
The most widely recognized nuisance variable for measures of metacognition is task performance[18]. The reason that task performance is a nuisance variable is that an ideal measure of metacognition should not be affected by whether a participant happens to be given an easier or a more difficult task. That is, the participant's estimated ability to provide informative confidence ratings should not change based on the difficulty of the object-level task that they are asked to perform. As mentioned earlier, this logic does not apply well to task manipulations, which is why I only examine stimulus manipulations here.

To quantify how task performance affects measures of metacognition, one needs datasets with multiple difficulty conditions and a large number of trials (either because of including many participants or many trials per participant). I selected three datasets from the Confidence Database that meet these characteristics: Shekhar (3 difficulty levels, 20 participants, 2800 trials/sub, 56,000 total trials), Rouault1 (70 difficulty levels, 466 participants, 210 trials/sub, 97,860

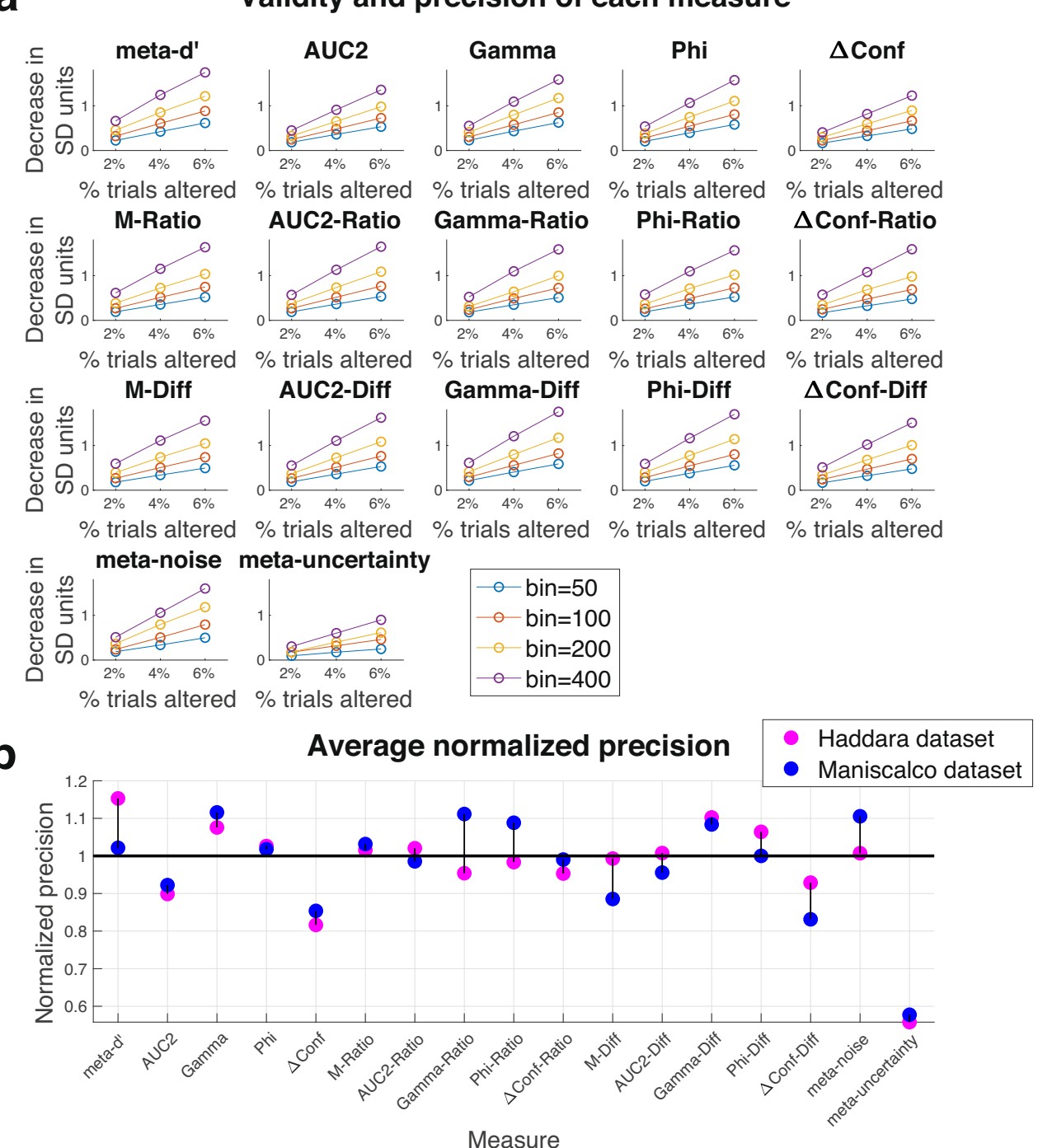

**Fig. 1 | Validity and precision of each measure.** Results of an artificial corruption of the confidence ratings where confidence for correct trials was decreased by 1, and confidence for incorrect trials was increased by 1. **a** Detailed results for the Haddara dataset (for detailed results on the Maniscalco dataset, see Supplementary Fig. 1). Each one of the 17 measures of metacognition showed a decrease with this manipulation. The plot shows the decrease in units of the standard deviation (SD) of the measure's fluctuations across different bins. The decrease was computed for bin sizes of 50, 100, 200, and 400 trials, as well as for 2, 4, and 6% of trials being corrupted. **b** Normalized precision for all 17 measures in each of the two datasets (Haddara and Maniscalco). The precision values are normalized such that the average precision level of the first 16 measures equals 1 in each of the two datasets. As can be seen, *meta-uncertainty* has substantially lower level of precision than the rest of the measures. The differences between the remaining measures are not always trivial but tend to be smaller.

total trials), and Rouault2 (many difficulty levels, 484 participants, 210 trials/sub, 101,640 total trials). Both Rouault datasets have a large range of difficulty levels which I split into low/high by taking a median split. I then computed each measure separately for each difficulty level and compared them using t-tests.

The results showed that all traditional measures that are not normalized in any way (i.e., *meta-d'*, *AUC2*, *Gamma*, *Phi*, and *ΔConf*) are strongly dependent on task performance: they all substantially increase as the task becomes easier ($p < 0.001$ for all five measures and three datasets; Fig. 2a; Supplementary Tables 3–5; see

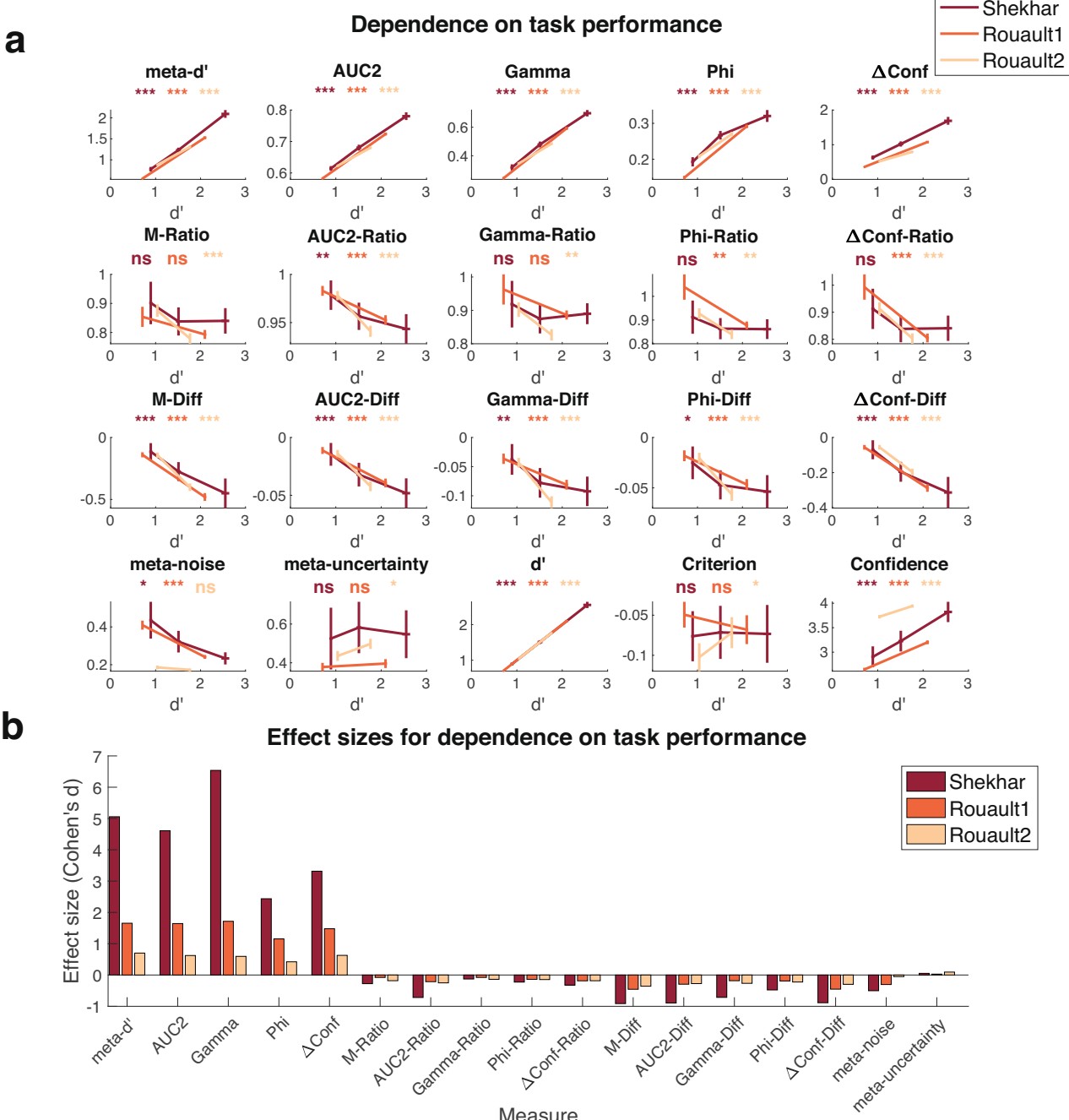

**Fig. 2 | Dependence of estimated metacognitive scores on task performance.**
**a** Estimated metacognitive ability for all 17 measures, as well as *d'*, criterion, and confidence for different difficulty levels in the Shekhar (*n* = 20), Rouault1 (*n* = 466), and Rouault2 (*n* = 484) datasets. Traditional measures of metacognition (top row) all showed a strong positive relationship with task performance, whereas all difference measures (third row) show a strong negative relationship. Ratio measures (second row) and the two model-based measures (meta-noise and meta-uncertainty) performed much better but still showed weak relationships with task performance. Error bars showing SEM are displayed on both the x and y axes. Statistical results are based on uncorrected two-sided t-tests comparing the highest to lowest difficulty level within each dataset for each measure (see Supplementary Tables 3–5 for complete results). ***, *p* < 0.001; **, *p* < 0.01; *, *p* < 0.05; ns, not significant. **b** Effect sizes for dependence on task performance. Effect size (Cohen's *d*) is plotted for each metric and dataset. As can be seen in the figure, non-normalized traditional measures (i.e., *meta-d', AUC2, Gamma, Phi,* and *ΔConf*) show strong positive relationship with task performance. Corrections with the ratio and difference methods reverse this relationship, with the ratio correction being clearly superior. The model-based metrics *meta-noise* and *meta-uncertainty* perform well too, with *meta-uncertainty* showing particularly low effect sizes.

Supplementary Fig. 2 for the same plots as a function of difficulty level instead of *d'* level). Critically, the increase across the five measures from the most difficult to the easiest had a very large effect size (Cohen's *d* = 2.47, 2.29, 2.95, 1.34, and 1.81 for each of the five measures after averaging across the four datasets; Fig. 2b).

Having established that these five measures strongly depend on task performance, I then examined whether normalizing them removes this dependence. The more popular method of normalization—the ratio method—indeed performed well. The average effect size (Cohen's *d*) for *M-Ratio, AUC2-Ratio, Gamma-Ratio, Phi-Ratio,* and *ΔConf-Ratio* was −0.18, −0.39, −0.11, −0.17, and −0.23, respectively.

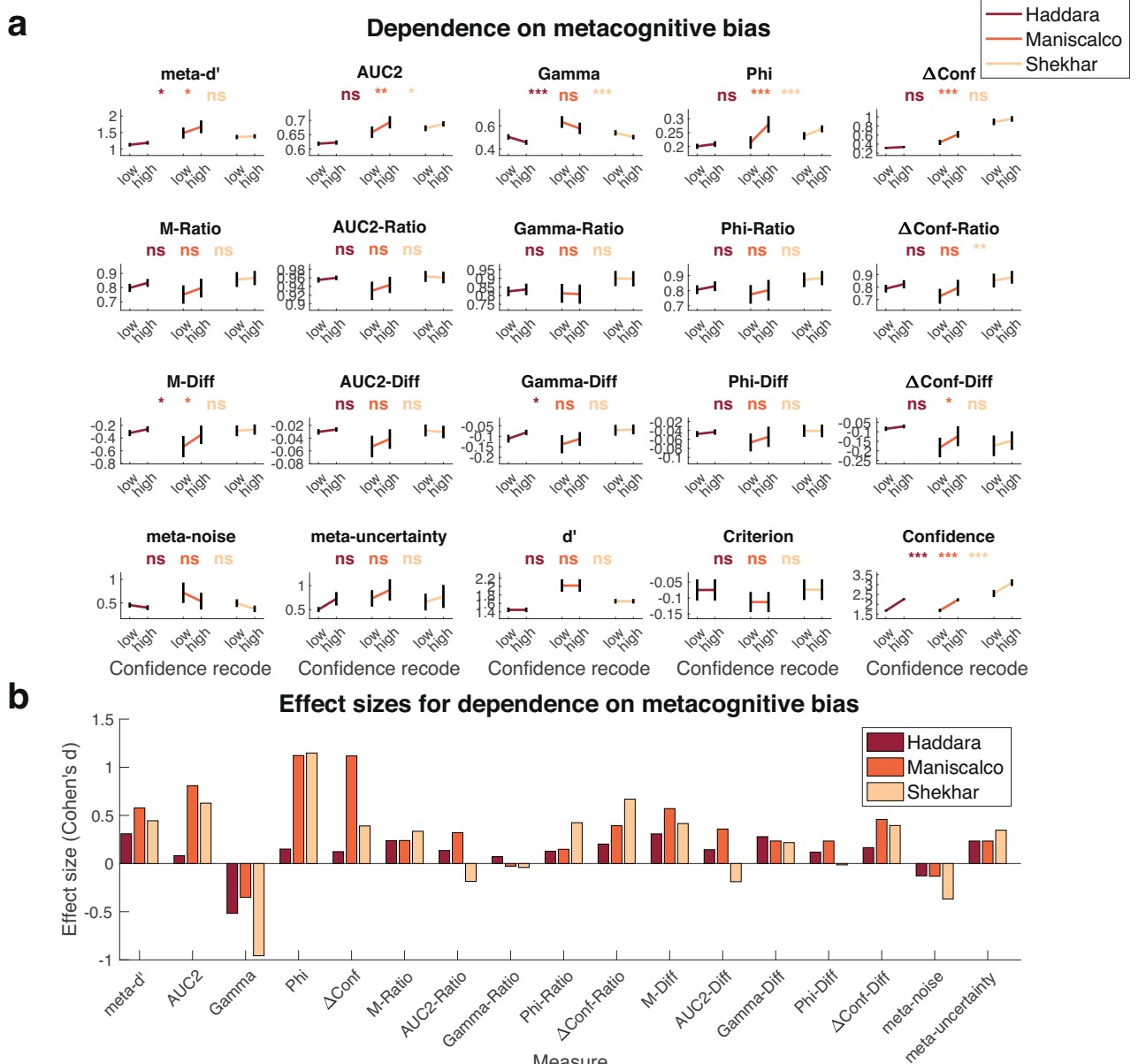

**Fig. 3 | Dependence of estimated metacognitive scores on metacognitive bias.** **a** Estimated metacognitive ability for all 17 measures, as well as *d′*, criterion, and confidence for data recoded to have lower or higher confidence in the Haddara ($n = 70$), Maniscalco ($n = 22$), and Shekhar ($n = 20$) datasets. Traditional measures of metacognition (top row) showed a medium-to-large positive relationship with metacognitive bias (except for *Gamma*, which showed a negative relationship). Ratio measures (second row) and the two model-based measures (meta-noise and meta-uncertainty) performed the best. Error bars show SEM. Statistical results are based on uncorrected two-sided t-tests comparing the high vs. low confidence recode within each dataset for each measure (see Supplementary Tables 6–8 for complete results). \*\*\*, $p < 0.001$; \*\*, $p < 0.01$; \*, $p < 0.05$; ns, not significant. **b** Effect sizes for dependence on metacognitive bias. Effect size (Cohen's *d*) is plotted for each metric and dataset. As can be seen in the figure, all metrics except for *Gamma* and *meta-noise* have a mostly positive relationship with metacognitive bias (i.e., higher confidence leads to higher estimates of metacognition). The smallest absolute effect sizes (under 0.15) occurred for *AUC2-Ratio*, *Gamma-Ratio*, *AUC2-Diff*, and *Phi-Diff*, but many other measures exhibited effect sizes in the small-to-medium range.

These are small effect sizes, except for *AUC2-Ratio* which has medium effect size. Nevertheless, it should be noted that the negative direction of the effect between task performance on metacognitive scores was consistent across all five measures and three datasets (with 9/15 tests being significant at $p < 0.05$; Supplementary Tables 3–5). Thus, while all ratio measures perform much better than the original metrics they are derived from, they tend to slightly overcorrect.

The five difference measures (*M-Diff, AUC2-Diff, Gamma-Diff, Phi-Diff,* and *ΔConf-Diff*) were much less effective in removing the dependence on task performance compared to their ratio counterparts. Indeed, they all exhibited an over-correction where easier conditions

led to lower scores with medium average Cohen's *d* effect sizes (*M-Diff*: −0.58; *AUC2-Diff*: −0.49; *Gamma-Diff*: −0.39; *Phi-Diff*: −0.30; *ΔConf-Diff*: −0.55). Further, the relationship between task performance and the metacognitive score was significantly negative for all five measures and three datasets ($p < 0.05$ for all 15 tests; Supplementary Tables 3–5). These results demonstrate that the difference measures uniformly fail at their main purpose, which is to remove the dependence of metacognitive measures on task performance.

Finally, the two model-based measures (*meta-noise* and *meta-uncertainty*) showed relatively weak but still systematic relationships with task difficulty. Specifically, *meta-noise* decreased for easier

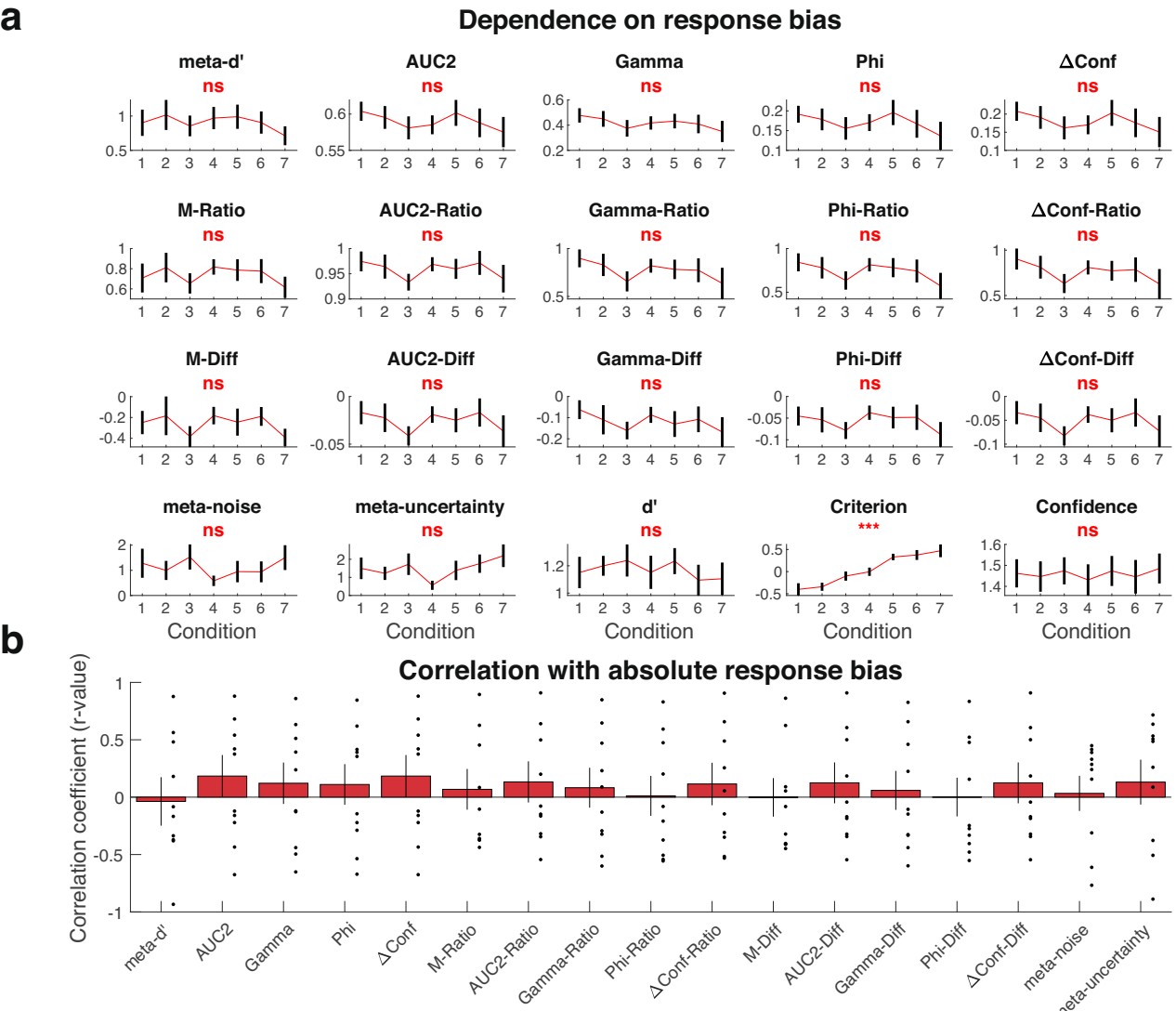

**Fig. 4 | Dependence of estimated metacognitive scores on response bias.**
**a** Estimated metacognitive ability for all 17 measures, as well as $d'$, criterion, and confidence for the seven conditions in the Locke ($n = 10$) dataset. As expected, the condition strongly affected response criterion, c. Despite that, condition did not significantly modulate any of the 17 measures of metacognition. The seven conditions in the graph are arranged based on their average criterion values. Error bars show SEM. Statistical results are based on repeated measures ANOVAs testing for the effect of condition on each measure (see Supplementary Table 9 for complete results). ***, $p < 0.001$; ns, not significant. **b** Correlation with absolute response bias. Average correlation between estimated metacognitive ability and absolute response bias (i.e., $|c|$) for all 17 measures ($n = 10$). As can be seen from the figure, all relationships are relatively small, but there is still a fair amount of uncertainty around each value. Error bars show SEM.

conditions in all three datasets (average Cohen's $d = -0.29$), whereas *meta-uncertainty* increased for easier conditions in all three datasets (average Cohen's $d = 0.06$). Both effects were associated with relatively small Cohen's $d$ effect sizes that were comparable to what was observed for the ratio measures. As such, both model-based measures perform as well as the ratio measures in controlling for task performance. Given that *meta-uncertainty* corrected in the opposite direction of the other viable measures (the ratio measures and *meta-noise*) and had the lowest absolute Cohen's $d$, studies that feature task performance confounds may benefit from performing analyses using both *meta-uncertainty* and at least one more measure.

### Dependence on metacognitive bias

A less appreciated nuisance variable is metacognitive bias: the tendency to give low or high confidence ratings for a given level of performance. Metacognitive bias can be measured simply as the average confidence in a condition. Recently, Shekhar and Rahnev[27] developed a method that involves recoding the original confidence ratings to examine how measures of metacognition depend on metacognitive bias. The method was further improved by Xue et al.[28]. The Xue et al. method consists of recoding confidence ratings as to artificially induce metacognitive bias toward lower or higher confidence ratings. Comparing the obtained values for a given measure of metacognition applied to the recoded confidence ratings allows us to evaluate whether the measure is independent of metacognitive bias.

Similar to quantifying precision, quantifying how metacognitive bias affects measures of metacognition requires datasets with very large number of trials coming from a single experimental condition. Consequently, I selected the same two datasets used to quantify precision since they have the largest number of trials per participant while also featuring a single experimental condition: Haddara (3000 trials per participant) and Maniscalco (1000 trials per participant). In addition, I also used the Shekhar dataset (3 difficulty levels, 2800 trials per participant) but analyzed each difficulty level in isolation and then

## Split-half reliability

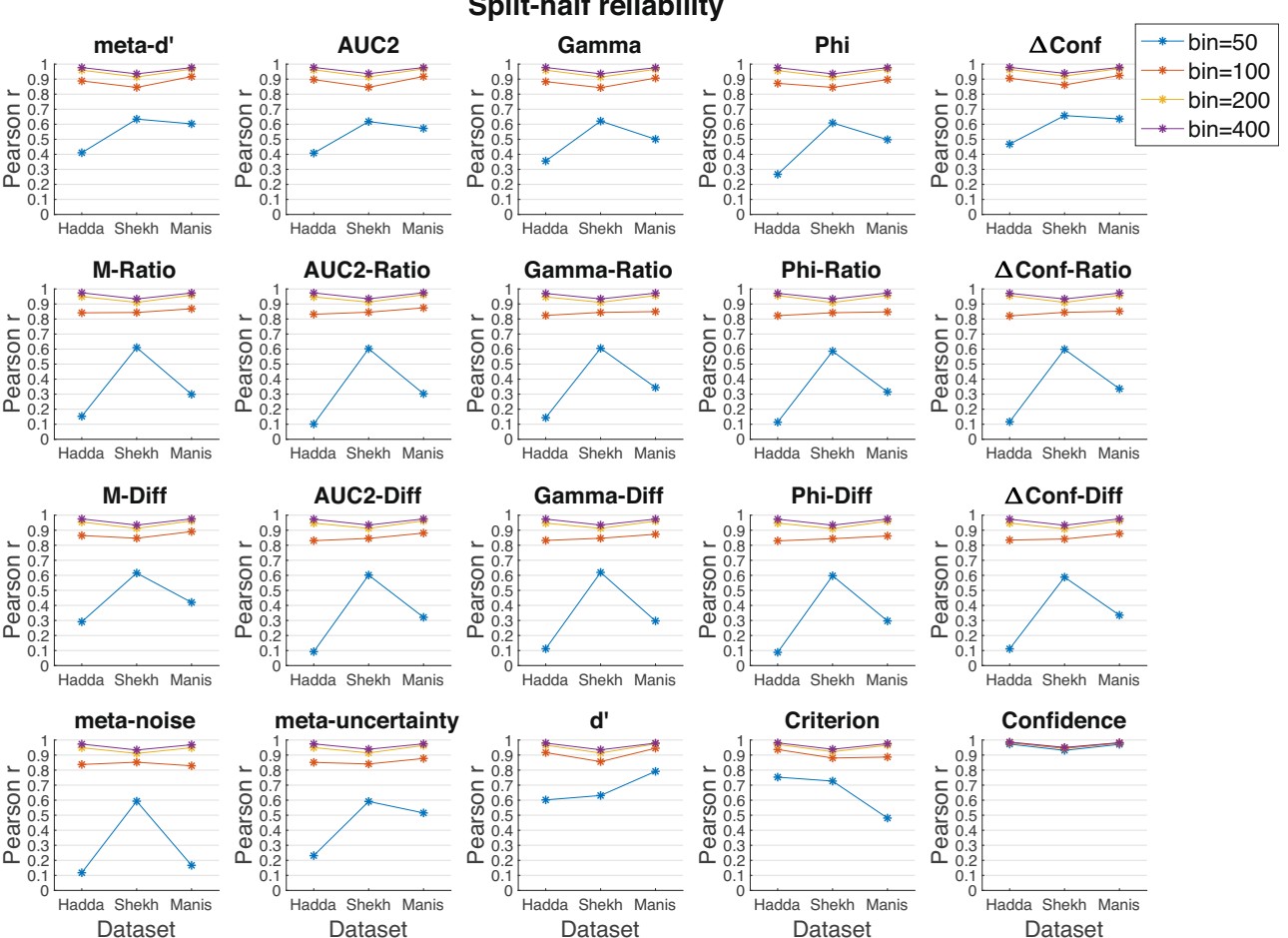

**Fig. 5 | Split-half reliability of metacognitive scores.** Correlations between each measure were computed based on odd vs. even trials for sample sizes of 50, 100, 200, and 400 trials. The figure shows that split-half correlations are high when at least 100 trials are used for computations but become unacceptably low when only 50 trials are used. The x-axis shows the results for three different datasets: Hadda (Haddara), Shekh (Shekhar), and Manis (Maniscalco).

averaged the results across the three difficulty levels. For that dataset, the continuous confidence scale was first binned into six levels as in the original publication[27].

The results demonstrated that *meta-d'*, *AUC2*, *Phi*, and *ΔConf* tend to increase with higher average confidence, whereas *Gamma* tends to decrease (Fig. 3a). The average (across the three datasets) Cohen's d effect size was in the medium-to-large range for all five measures (*meta-d'*: 0.44; *AUC2*: 0.51; *Gamma*: −0.61; *Phi*: 0.81; *ΔConf*: 0.54; Fig. 3b). In other words, all five non-normalized measures of meta-cognition depend on metacognitive bias. All five ratio measures had a positive relationship with metacognitive bias but with smaller Cohen's d effect sizes (*M-Ratio*: 0.27; *AUC2-Ratio*: 0.09; *Gamma-Ratio*: 0.001; *Phi-Ratio*: 0.23; *ΔConf-Ratio*: 0.42). Difference measures performed similarly to ratio measures (*M-Diff*: 0.43; *AUC2-Diff*: 0.10; *Gamma-Diff*: 0.24; *Phi-Diff*: 0.11; *ΔConf-Diff*: 0.34). Finally, the two model-based measures performed similar to the ratio and difference measures and exhibited low-to-medium effect sizes that again went in opposite directions of each other (*meta-noise*: −0.21; *meta-uncertainty*: 0.27). Note that the scores after recoding were similar but slightly larger than the original metacognitive scores before recoding (Supplementary Fig. 3). Overall, researchers who want to control for metacognitive bias would appear to do best if they used *AUC2-Ratio*, *Gamma-Ratio*, *AUC2-Diff*, or *Phi-Diff* as these all featured absolute effect sizes under 0.15. Nevertheless, given that *meta-noise* corrected in the opposite direction of the ratio and difference measures, it may be advisable for results obtained using one of those metrics to be reproduced with *meta-noise*.

### Dependence on response bias

The final nuisance variable examined here is response bias. Response bias can be measured simply as the decision criterion *c* in signal detection theory. To understand how response bias affects measures of metacognition, one needs datasets where the response criterion is experimentally manipulated and confidence ratings are simultaneously collected. Very few such datasets exist and only a single such dataset is featured in the Confidence Database. The dataset—named here Locke[44]—features seven conditions with manipulations of both prior and reward. Rewards were manipulated by changing the payoff for correctly choosing category 1 vs. category 2 (e.g., R = 4:2 means that 4 vs. 2 points were given for correctly identifying categories 1 and 2, respectively), whereas priors were manipulated by informing participants about the probability of category 2 (e.g., P = 0.75 means that there was 75% probability of presenting category 2 and 25% probability of presenting category 1). The seven categories were as follows (1) P = 0.5, R = 3:3, (2) P = 0.75, R = 3:3, (3) P = 0.25, R = 3:3, (4) P = 0.5, R = 4:2, (5) P = 0.5, R = 2:4, (6) P = 0.75, R = 2:4, and (7) P = 0.25, R = 4:2. The Locke dataset included many trials per condition (700) but relatively few participants (N = 10) and collected confidence on a 2-point scale.

The results suggested that none of the 17 measures of metacognition are strongly influenced by response bias (Fig. 4a). Indeed, while a repeated measures ANOVA revealed a very strong effect of condition on response criterion ($F(6,54) = 12.18$, $p < 0.001$, $\eta_p^2 = 0.58$), it showed no significant effect of condition on any of the measures of metacognition (all $p$'s > 0.13 for 17 tests; Supplementary Table 9). Critically, I computed

the correlation between the estimated metacognitive ability for each of the 17 measures and the absolute value of the response criterion (i.e., $|c|$). The idea behind this analysis is to investigate whether more extreme response bias (either positive or negative) is associated with increases or decreases in estimated metacognitive ability. The results demonstrated that all correlation coefficients were very small (all r-values were between −0.04 and 0.21; Fig. 4b). There was a fair amount of uncertainty about these values, as seen by the wide error bars in Fig. 4b, so it is possible some of these relationships may be stronger than the current data suggest. Overall, these results should be interpreted with caution given the small sample size and the fact that a 2-point confidence scale may be noisier for estimating metacognitive scores. Nonetheless, these initial findings suggest that response bias may not have a large biasing effect on measures of metacognition.

## Reliability

Measures of metacognition are often used in studies of individual differences to examine across-participant correlations between metacognitive ability and many different factors such as brain activity and structure[10,11,50], metacognitive ability in other domains[51,52], psychiatric symptom dimensions[46], cognitive processes such as confidence leak[12], etc. These types of studies require measures of metacognition to have high reliability. (Note that the reliability of a measure is enhanced by both high precision and large spread of scores across participants, so both of these two factors are important for between-subject analyses. In contrast, within-subject analyses only require high precision. Therefore, low-reliability scores are not necessarily problematic for within-subject designs.)

Perhaps surprisingly, relatively little has been done to quantify the reliability of measures of metacognition (but see refs. 14,41). Here I examine split-half reliability (correlation between estimates obtained from odd vs. even trials) and test-retest reliability (correlation between estimates obtained on different days).

### Split-half reliability

To examine split-half reliability for different sample sizes, one needs datasets with many trials per participant and a single condition (or large number of trials per condition if multiple conditions are present). Consequently, I selected the same three datasets used to examine the dependence of measures of metacognition on metacognitive bias: Haddara (3000 trials per participant), Maniscalco (1000 trials per participant), and Shekhar (3 difficulty levels, 2800 trials per participant). As before, I analyzed each difficulty level in the Shekhar dataset in isolation and then averaged the results across the three difficulty levels. For each dataset, I computed each measure of metacognition based on odd and even trials separately and correlated the two. To examine how split-half reliability depends on sample size, I performed the procedure above for bins of 50, 100, 200, and 400 trials separately. Because the datasets contained multiple bins of each size, I averaged the results across all bins of a given size.

The results showed that measures of metacognition have good split-half reliability as long as the measures are computed using at least 100 trials (Fig. 5). Indeed, bin sizes of 100 trials produced split-half correlations of $r > 0.837$ for all 17 measures when averaged across the three datasets with an average split-half correlation of $r = 0.861$. These numbers increased further for bin sizes of 200 (all $r$'s > 0.938, average $r = 0.946$) and 400 trials (all $r$'s > 0.961, average $r = 0.965$). Further, these numbers were only a little lower than the split-half correlations for $d'$ (100 trials: $r = 0.913$; 200 trials: $r = 0.958$; 400 trials: $r = 0.970$). However, the split-half correlations strongly diminished when the measures of metacognition were computed based on 50 trials with an average $r = 0.424$ and no measure exceeding $r = 0.6$. It should be noted that while performing better, $d'$ also had a relatively low split-half reliability of $r = 0.685$ when computed based on 50 trials. These results

suggest that individual difference studies should employ 100 trials per participant at a minimum and that there is little benefit in terms of split-half reliability for using more than 200 trials.

### Test-retest reliability

Split-half reliability is a useful measure of the intrinsic noise present in the across-subject correlations that can be expected in studies of individual differences. However, they do not account for fluctuations that could occur from day to day. These fluctuations can be examined by computing measures of metacognition obtained from different days, thus estimating what is known as test-retest reliability. Such estimation requires datasets with multiple days of testing and a large number of trials per participant per day. Only one dataset in the Confidence Database meets these criteria: Haddara (6 days; 3000 total trials per participant; 70 participants). I examined test–retest reliability by computing both intraclass correlation (ICC) and Pearson correlation between all pairs of days and then averaged across the different pairs.

The results showed very low test–retest reliability values (Fig. 6). Even with 400 trials used for estimation, no measure of metacognition exceeded an average ICC reliability of 0.75 and none of the measures outside of the five non-normalized and non-model-based measures (i.e., meta-d', AUC2, Gamma, Phi, and ΔConf) reached ICC reliability of 0.5, which is often considered the threshold for poor reliability. For example, the widely used measure M-Ratio had average ICC reliability of $r = 0.16$ (for 50 trials), 0.23 (for 100 trials), 0.29 (for 200 trials), and 0.42 (for 400 trials). The measure with highest test–retest correlation was ΔConf with ICC reliability of 0.39 (for 50 trials), 0.53 (for 100 trials), 0.65 (for 200 trials), and 0.75 (for 400 trials). Notably, test-retest reliability was not much higher for $d'$ or criterion $c$ compared to ΔConf (average difference of about 0.1) and was only robustly high for confidence (above 0.86 regardless of sample size). Similar test–retest correlation coefficients were obtained when Pearson correlation was computed instead of ICC (Fig. 6). These results are in line with the findings of Kopcanova et al.[14] and suggest that correlations between measures of metacognition and measures that do not substantially fluctuate on a day-by-day basis (e.g., structural brain measures) are likely to be particularly noisy such that very large sample sizes may be needed to find reliable results.

### Across-subject correlations between different measures

Lastly, I examined how different measures are related to each other by performing across-subject correlations. Note that these analyses should be interpreted with extreme caution because the correlation between two measures could be driven by a third factor. For these analyses, I again used the Haddara (3000 trials per participant), Maniscalco (1000 trials per participant), and Shekhar (3 difficulty levels, 2800 trials per participant) datasets. As in previous analyses, I examined each difficulty level in the Shekhar dataset in isolation and then averaged the results across the three difficulty levels. For each dataset, I computed each measure of metacognition based on all trials in the experiment and examined the across-subject correlations between different measures.

Overall, the 17 measures of metacognition showed medium-sized across-subject correlations with each other (average $r = 0.49$, 0.55, and 0.56 for the Haddara, Maniscalco, and Shekhar datasets, respectively; Supplementary Fig. 4). These analyses seemed to reveal three groups of measures. The first group consists of the five non-normalized measures (meta-d', AUC2, Gamma, Phi, and ΔConf), which exhibited average inter-measures correlation of 0.60 ($r = 0.60$, 0.63, and 0.58 in each dataset). The second group consists of the five ratio and five difference measures, which exhibited average inter-measures correlation of 0.63 ($r = 0.62$, 0.62, and 0.63 in each dataset). The average correlation between the first two groups of measures was slightly weaker than the within-group correlations ($r = 0.51$ on average;

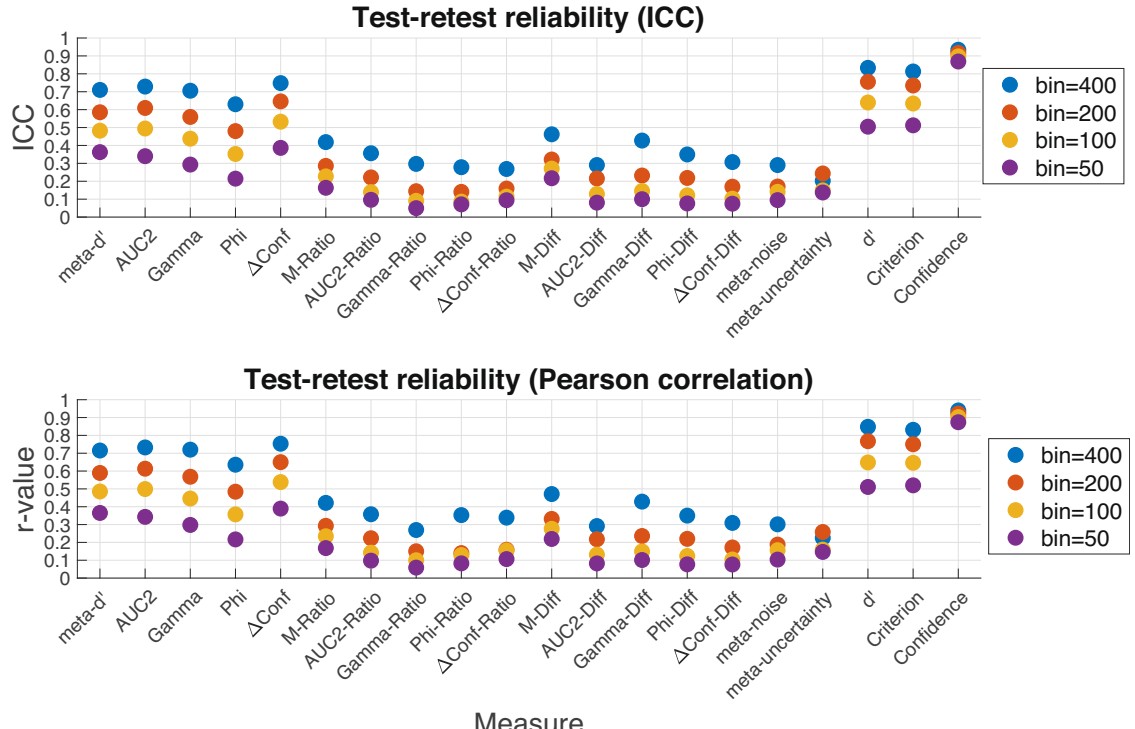

**Fig. 6 | Test–retest reliability of metacognitive scores.** Test–retest correlations in the Haddara dataset (6 days, 500 trials per day, 70 participants) show generally low test-retest reliability. Upper panel shows ICC values, whereas lower panel shows Pearson correlation. The test–retest reliability was low-to-moderate for the measures *meta-d'*, *AUC2*, *Gamma*, *Phi*, and *ΔConf* and very low for the remaining measures.

$r = 0.42$, 0.55, and 0.55 in each dataset). Note that these results could be driven by the fact that all five non-normalized measures are strongly driven by d', thus increasing the correlations between them. It may also be that the SDT-based normalization makes all ratios and difference measures similar to each other.

Finally, the third group of measures consists of the two model-based measures, which showed the strongest divergence from the rest of the measures. Specifically, *meta-noise* had an average correlation of 0.35 with the remaining measures ($r = 0.35$, 0.34, and 0.37 in each dataset) and *meta-uncertainty* had an average correlation of 0.44 with the remaining measures ($r = 0.33$, 0.45, and 0.53 in each dataset). The measures *meta-noise* and *meta-uncertainty* had a very weak correlation with each other ($r = 0.15$, 0.03, and 0.06 in each dataset). These results suggest that the two model-based measures may capture unique variance related to metacognitive ability.

## Discussion

Despite substantial interest in developing good measures of metacognition, there has been surprisingly little empirical work into the psychometric properties of current measures. Here I investigate the properties of 17 measures of metacognition, including eight new variants. I develop a method of determining the validity and precision of a measure of metacognition and examine each measure's dependence on nuisance variables and its split-half and test-retest reliability. The results paint a complex picture. No measure of metacognition is "perfect" in the sense of having the best psychometric properties across all criteria. Researchers need to make informed decisions about which measures to use based on the empirical properties of the different measures. The results are summarized in Fig. 7.

### Validity and precision

I found that all 17 measures of metacognition examined here are valid. With the exception of *meta-uncertainty*, all measures seem to have comparable level of precision. This result is rather surprising and

suggests that precision may be limited by measurement error such that it is unlikely that any new measure of metacognition can substantially exceed the precision level found for the first 16 measures here. Nevertheless, new measures can be noisier and therefore it is critical to demonstrate their level of precision. Note that less precise measures can also appear to depend less on nuisance factors not because of their better psychometric properties but due to their noisiness.

### Dependence on task performance

Task performance is arguably the most important and best-appreciated nuisance variable for measures of metacognition. As has been previously suspected[18], the results here show that all traditional measures of metacognition are strongly dependent on task performance. However, the ratio method does a very good job of correcting for this dependence with *M-Ratio*, *Gamma-Ratio*, *Phi-Ratio*, and *ΔConf-Ratio* showing only weak dependence on task performance. On the other hand, the difference method performed poorly in removing the dependence on task performance. The model-based measures *meta-noise* and *meta-uncertainty* also performed well.

### Dependence on metacognitive bias

Previous research has shown that *meta-d'* and *M-Ratio* are positively correlated with metacognitive bias such that a bias toward higher confidence also leads to high values for these measures[27,28]. The current investigation replicated these previous results and showed that similar effects are observed for many other measures. Nevertheless, the dependence was of low to medium effect size for *M-Ratio* and comparable to newer measures such as *meta-noise* and *meta-uncertainty*.

### Dependence on response bias

The results for response bias should be considered preliminary because they are based on a single dataset that consists of 10 participants. As such, the results should not be taken as strong evidence for an absence of dependence on response bias (hence, all measures are

| Measure | Precision | Dependence on task performance | Dependence on metacognitive bias | Dependence on response bias | Split-half reliability | Test-retest reliability | Unique limitations | Unique advantages |
|---|---|---|---|---|---|---|---|---|
| meta-d' | Pr = .65 | d = 2.47 | d = 0.44 | r = -.04 | r = .89 | ICC = .71 | | |
| AUC2 | Pr = .54 | d = 2.29 | d = 0.51 | r = .18 | r = .89 | ICC = .73 | | Continuous |
| Gamma | Pr = .65 | d = 2.95 | d = -0.61 | r = .12 | r = .88 | ICC = .71 | | Continuous |
| Phi | Pr = .61 | d = 1.34 | d = 0.81 | r = .11 | r = .87 | ICC = .63 | | Continuous |
| ΔConf | Pr = .50 | d = 1.81 | d = 0.54 | r = .18 | r = .90 | ICC = .75 | | Continuous |
| M-Ratio | Pr = .61 | d = -0.18 | d = 0.27 | r = .07 | r = .85 | ICC = .42 | Unstable for low d' | |
| AUC2-Ratio | Pr = .60 | d = -0.39 | d = 0.09 | r = .13 | r = .85 | ICC = .36 | | |
| Gamma-Ratio | Pr = .61 | d = -0.11 | d = 0.001 | r = .08 | r = .84 | ICC = .30 | Unstable for low d' | |
| Phi-Ratio | Pr = .62 | d = -0.17 | d = 0.23 | r = .01 | r = .84 | ICC = .28 | Unstable for low d' | |
| ΔConf-Ratio | Pr = .58 | d = -0.23 | d = 0.42 | r = .11 | r = .84 | ICC = .27 | Unstable for low d' | |
| M-Diff | Pr = .56 | d = -0.58 | d = 0.43 | r = -.002 | r = .87 | ICC = .47 | | |
| AUC2-Diff | Pr = .59 | d = -0.49 | d = 0.10 | r = .12 | r = .85 | ICC = .29 | | |
| Gamma-Diff | Pr = .65 | d = -0.39 | d = 0.24 | r = .06 | r = .85 | ICC = .43 | | |
| Phi-Diff | Pr = .62 | d = -0.30 | d = 0.11 | r = .001 | r = .85 | ICC = .35 | | |
| ΔConf-Diff | Pr = .53 | d = -0.55 | d = 0.34 | r = .12 | r = .85 | ICC = .31 | | |
| meta-noise | Pr = .63 | d = -0.29 | d = -0.21 | r = .03 | r = .84 | ICC = .29 | Cannot be negative | Model-based |
| meta-uncertainty | Pr = .34 | d = 0.06 | d = 0.27 | r = .13 | r = .86 | ICC = .21 | Cannot be negative | Model-based |

**Fig. 7 | Summary of results.** The figure lists the values obtained for each measure of metacognition for various criteria. Precision is the measure developed in this paper and the values listed are the average of the values in Fig. 1b, c. Higher precision values are better. For dependence of task performance and metacognitive bias, the figure lists the average Cohen's *d* values reported in the paper. For dependence on response bias, the figure lists the average correlation between each measure of metacognition and the absolute value of response bias ($|c|$). Lower absolute value of these dependencies is better. The reported split-half reliability is the average value across datasets obtained for a bin size of 100, whereas the reported test-retest reliability (ICC) is the average value obtained for a bin size of 400. Higher reliability values are better. Color coding is meant as a general indicator but should be interpreted with caution. Green indicates very good properties, yellow indicates good properties, orange indicates problematic properties, and red indicates bad properties. Colors were assigned based on the following thresholds: 0.5 for precision, 0.3 and 1 for Cohen's *d*, 0.5 for test–retest reliability. Green was not used in any of the columns regarding dependence on nuisance variables as to not give the impression that any measure is certainly independent of any of the nuisance variables. The figure also lists several unique advantages and disadvantages of each measure discussed in the main text.

colored in yellow rather than green in Fig. 7). Yet, it does appear that any dependencies are unlikely to be particularly strong, at least for the range of response bias likely to occur in most experiments.

## Alternative ways of quantifying dependence on nuisance variables

I quantified the dependence on nuisance factors by examining effect sizes (Cohen's *d* and *r*-values). Alternative ways of examining the dependence on nuisance variables make it difficult to compare the measures. For example, the difference or ratio of the raw values across easy vs. difficult conditions is not readily comparable across metrics that take different ranges. The main limitation of the approach I adopted (examining effect sizes) is that noisier measures will have an advantage. In practice, the precision analysis found that 16 of the 17 measures examined here have a similar level of precision, and thus do not substantially differ in their noisiness. Nevertheless, it is possible that the relatively low dependence of *meta-uncertainty* on nuisance variables is in part due to its lower precision (higher noisiness).

## Split-half reliability

Guggenmos[41] recently examined many datasets in the Confidence Database and concluded that split-half reliability for *M-Ratio* is relatively poor (*r* ~ 0.7 for bin sizes between 400 and 600). (Note that the paper computes split-half reliability but it calls it test-retest reliability.) One issue with the approach by Guggenmos is that many of the analyzed datasets in the Confidence Database feature a variety of conditions, manipulations, and sample sizes. These factors may reduce the observed split-half reliability. Indeed, focusing on a select number of large datasets with a single condition at a time, the current paper finds much higher split-half reliabilities (between 0.84 and 0.9

for a bin size of 100). These results suggest that for sample sizes of 100 or more, one can expect reliable estimates of metacognition for every measure when using a single experimental condition. It is likely that studies that mix different conditions and estimate metacognitive scores across all of them would produce lower split-half reliability in line with the results of Guggenmos. Note that sample sizes of 50 produced unacceptably low reliabilities, so 100 should be considered as a rough lower bound for the necessary number of trials when estimating metacognition in studies of individual differences.

## Test–retest reliability

One of the most striking results here is the very low test-retest reliabilities observed. Besides the five non-normalized measures (*meta-d'*, *AUC2*, *Gamma*, *Phi*, and *ΔConf*), no other measure showed test-retest reliability exceeding 0.5 even for sample sizes of 400 trials. However, the non-normalized five measures are strongly dependent on task performance, and thus their higher reliability may be partly (or wholly) due to the higher reliability of task performance itself (test-retest reliability of *d'* was 0.84 for a sample size of 400). Therefore, studies that match *d'* for all participants may result in test-retest reliability values for these five measures of metacognition that are as low as the remaining measures. Nevertheless, these results are based on a single dataset and should therefore be replicated before strong recommendations can be made. That said, the results are consistent with a recent paper that examined the test-retest reliability of *M-Ratio* in a sample of 25 participants[14]. Therefore, researchers who study individual differences in metacognition should be aware of the potential low test-retest reliability of measures of metacognition, which may explain previous failures to find significant correlations between metacognitive abilities across domains.

## Unique advantages and disadvantages of different measures

Several measures feature unique advantages and disadvantages (Fig. 7). For example, four of the ratio measures (*M-Ratio*, *Gamma-Ratio*, *Phi-Ratio*, and *ΔConf-Ratio*) become unstable for difficult conditions because they include division by variables ($d'$, expected Gamma, expected Phi, and expected ΔConf, respectively) that are very close to 0 in such conditions. These measures should therefore be used preferentially when performance levels are relatively high (e.g., one should aim for $d'$ values above 1, which roughly corresponds to accuracy values above 69%).

An advantage of *AUC2*, *Gamma*, *Phi*, and *ΔConf* is that they all work well with continuous confidence scales. All other measures rely on SDT-based computations that necessitate that continuous scales are binned before analyses. Such binning may lead to loss of information, but it is currently unclear how much signal may be lost by different binning methods.

The two model-based measures—*meta-noise* and *meta-uncertainty*—have unique advantages and disadvantages. Their main advantage is that all their underlying assumptions are explicitly known. Conversely, other measures must necessarily include hidden assumptions that are difficult to reveal without linking them to a process model of metacognition[3]. Another unique advantage of these measures is that they can in principle be applied much more flexibly. For example, when an experiment contains several conditions, other measures do not allow the estimation of a single measure of metacognition and simply ignoring the different conditions can lead to inflated scores[49]. Conversely, both *meta-noise* and *meta-uncertainty* allow different conditions to be modeled as part of their underlying process models and thus a single metacognitive score can be computed in a principled way across many conditions. A possible disadvantage of both measures is that they can only take positive values and therefore cannot be used for situations where metacognition may contain more information than the decision itself, such as in the presence of additional information that arrives after the decision[53,54].

Several measures showed dependence on nuisance variables that went in the opposite direction from most other measures (*meta-uncertainty* for task performance, as well as *meta-noise* and *Gamma* for metacognitive bias). As such, these measures may be especially useful to use when there is a concern that results may be driven by a specific nuisance variable. Unfortunately, it is currently difficult to determine why these measures show the opposite effects (or, for that matter, why most measures show the dependencies they show). Understanding the nature of these relationships will likely require further progress in developing well-fitting process models of metacognition[55,56].

## Is *M-Ratio* still the gold standard for measuring metacognition?

In the last decade, *M-Ratio* has become the dominant measure of metacognition due to its assumed better psychometric properties[18,34,57]. This status has naturally attracted greater scrutiny and many recent papers have criticized some of the properties of *M-Ratio*[27,28,37,41,58]. However, while these criticisms are valid, papers have rarely tested how alternative measures perform on the same tests. The results here demonstrate that across all examined dimensions, there are no measures that clearly outperform *M-Ratio*. Three measures—*meta-noise*, *Gamma-Ratio*, and *Phi-Ratio*—showed very similar performance to *M-Ratio*, while all other measures appear inferior to *M-Ratio* in at least one critical dimension: they strongly depend on task performance (all five non-normalized measures, all five difference measures, and *AUC-Ratio*), have low precision (*meta-uncertainty*), or strong dependence on metacognitive bias (*ΔConf-Ratio*). I see no strong argument in the present data to choose either *Gamma-Ratio* or *Phi-Ratio* over *M-Ratio*, especially given how established *M-Ratio* is contrary to *Gamma-Ratio* and *Phi-Ratio*. There are good arguments for using *meta-noise* in addition to *M-Ratio* as a way of controlling for metacognitive bias given that the two measures depend on

metacognitive bias in opposite directions. Similarly, *meta-uncertainty* can also be used in addition to *M-Ratio* or *meta-noise* to control for task performance given that it depends on task performance in the opposite direction than the other two measures.

There are strong reasons for the field to transition to model-based measures of metacognition[3] since model-based measures are uniquely positioned to properly capture the influence of metacognitive inefficiencies[59]. The measure *meta-noise* is especially promising given its good performance on the current tests and the fact that its associated model is a successful model of metacognition[55]. That said, *meta-noise* is currently only implemented in Matlab (see codes associated with the current paper) and is more computationally intensive. Thus, although *meta-noise* or other model-based measures of metacognition should eventually supplant *M-Ratio*, for the time being it is hard to justify abandoning *M-Ratio* as the gold standard for the field.

## Limitations

The present work has several limitations. First, despite the attempt to be comprehensive, several measures of metacognition have been omitted including recent model-based measures[30,60], different variants of *M-Ratio*[41], and legacy measures such as Kunimoto's $a$[58]. Nevertheless, the current work should make it much easier for researchers to establish the properties of other measures of metacognition and compare them to the ones examined here. Second, while I have attempted to use multiple large datasets for each analysis, two of the analyses only included a single dataset (dependence on response bias and test-retest reliability) and should be interpreted with caution. Even in cases where multiple datasets were used, it is clear that adding more datasets would alter the values in Fig. 7. As such, the values there should be understood as rough estimates that are bound to be improved upon by future work that analyzes additional large datasets. Third, all ratio and difference measures were computed using SDT with equal variance; computations assuming unequal variance may lead to different results. Fourth, the current analyses were conducted exclusively in the context of perception. Metacognition has been widely studied in the context of learning, memory, problem solving, etc[1]. While the results here are expected to generalize to these other domains, additional research is needed to confirm that. Fifth, most measures examined here only apply to 2-choice tasks and thus cannot be used for designs with estimation tasks, n-choice tasks, etc.

## Recommendations

Based on the current set of results and findings from the greater literature, Table 3 lists recommendations for researchers interested in measuring metacognitive ability. The recommendations pertain to experimental design, analysis, and interpretation.

Researchers interested in measuring metacognition precisely need to pay special attention to experimental design. They should use relatively easy tasks (while still avoiding ceiling effects) because ratio measures become unstable for low $d'$. They should also ideally use a single difficulty level to avoid the inflation that arises when multiple difficulty levels are combined[49]. Finally, researchers need to ensure adequate sample sizes. I recommend at least 400 trials per participant for individual differences research and at least 100 trials per participant for within-subject studies.

At the level of analysis, I recommend using more than one measure whenever possible, especially if the results could plausibly depend on task performance or metacognitive bias. Difference measures should not be assumed to properly correct for task performance. In cases where performance is very low and ratio measures are unstable, the results should be confirmed by examining both difference and non-normalized measures (since these two categories have opposite dependence on task performance). When multiple conditions are present, researchers should ideally use the model-based measures *meta-noise* or *meta-uncertainty* via custom modeling.

**Table 3 | Recommendations for metacognition researchers**

| Recommendations | |
| --- | --- |
| Experimental design | 1. Use relatively easy tasks to avoid instability related to low $d'$ values<br>2. Whenever possible, use designs with a single difficulty level<br>3. Collect at least 100 trials per participant<br>4. For individual differences research, ideally collect at least 400 trials per participant |
| Analysis | 1. Use several measures of metacognition<br>2. *M-Ratio* continues to be a good default measure of metacognitive ability<br>3. If results could plausibly depend on task performance or metacognitive bias, then confirm that results remain the same when using *meta-noise* or *meta-uncertainty*<br>4. Do not use difference measures to correct for differences in task performance<br>5. If multiple conditions are present, use *meta-noise* or *meta-uncertainty* (custom modeling necessary) |
| Interpretation | 1. Do not automatically assume that *M-Ratio* < 1 indicates signal loss from the decision to the metacognitive system<br>2. Do not automatically assume that *M-Ratio* > 1 indicates signal gain from the decision to the metacognitive system |

Finally, researchers should interpret findings of *M-Ratio* and other ratio measures being larger (or smaller) than 1 with caution. Traditionally, such findings have been interpreted as the metacognitive system having more (or less) signal than the decision-making system. However, many other factors can drive such results, such as the mixing of several difficulty levels[49] or criterion (as opposed to signal) noise[59], and some researchers have even questioned the separation of decision-making and metacognitive systems[61].

## Methods

### Ethical regulations

The current study complies with all relevant ethical regulations. All analyses were performed on deidentified data from publicly available datasets and thus were exempt from Internal Review Board review.

### Datasets

To investigate the empirical properties of measures of metacognition, I used the datasets from the Confidence Database[47] that are most appropriate for each individual analysis. This process resulted in the selection of six different datasets, briefly discussed below in alphabetical order. In each case, participants completed a 2-choice perceptual task and provided confidence ratings. For each dataset, I only considered trials from the main experiment and removed any staircase or practice trials that may have been included. In addition, I excluded participants who had lower than 60% or higher than 95% accuracy, or who gave the same object-level or confidence response on more than 85% of trials. These exclusions were made because such participants can have unstable metacognitive scores. Overall, these criteria led to the exclusion of 58 out of 1091 participants (5.32% exclusion rate). Data were collected in a lab setting unless otherwise indicated.

### Haddara dataset

The first dataset is named "Haddara_2022_Expt2" in the Confidence Database (simplified to "Haddara" here) and consists of 75 participants each completing 3350 trials over seven days. Because Day 1 consisted of a smaller number of trials (350) compared to Days 2–7 (500 trials each), I only analyzed the data from Days 2–7 (3000 trials total). All experimental details can be found in the original publication[43]. Briefly, the task was to determine the more frequent letter in a 7 × 7 display of X'es and O's. Confidence was provided on a 4-point scale using a separate button press. The data collection was conducted online and half the participants received trial-by-trial feedback (all participants are considered jointly here). Five participants were excluded from this dataset (6.67% exclusion rate).

### Locke dataset

The second dataset is named "Locke_2020" in the Confidence Database (simplified to "Locke" here) and consists of 10 participants each completing 4900 trials. All experimental details can be found in the original publication[44]. Briefly, the task was to determine if a Gabor patch was tilted to the left or right vertical. Confidence was provided on a 2-point scale using a separate button press. There were seven conditions with manipulations of both prior and reward. Rewards were manipulated by changing the payoff for correctly choosing category 1 vs. category 2 (e.g., $R = 4:2$ means that 4 vs. 2 points were given for correctly identifying categories 1 and 2, respectively), whereas priors were manipulated by informing participants about the probability of category 2 (e.g., $P = 0.75$ means that there was 75% probability of presenting category 2 and 25% probability of presenting category 1). The seven categories were as follows (1) $P = 0.5$, $R = 3:3$, (2) $P = 0.75$, $R = 3:3$, (3) $P = 0.25$, $R = 3:3$, (4) $P = 0.5$, $R = 4:2$, (5) $P = 0.5$, $R = 2:4$, (6) $P = 0.75$, $R = 2:4$, and (7) $P = 0.25$, $R = 4:2$. There were equal number of trials (700) per condition. No participants were excluded from this dataset.

### Maniscalco dataset

The third dataset is named "Maniscalco_2017_expt1" in the Confidence Database (simplified to "Maniscalco" here) and consists of 30 participants each completing 1000 trials. All experimental details can be found in the original publication[45]. Briefly, the task was to determine which of the two patches presented to the left and right of fixation contained a grating. A single difficulty condition was used throughout. Confidence was provided on a 4-point scale using a separate button press. Eight participants were excluded from this dataset (26.67% exclusion rate).

### Rouault1 and Rouault2 datasets

The fourth and fifth datasets are named "Rouault_2018_Expt1" and "Rouault_2018_Expt2" in the Confidence Database (simplified to "Rouault1" and "Rouault2" here). They consist of 498 and 497 participants, respectively, each completing 210 trials. All experimental details can be found in the original publication that describes both datasets[46]. Briefly, the task was to determine which of the two squares presented to the left and right of fixation contained more dots and then rate confidence using a separate button press. The Rouault1 dataset had 70 difficulty conditions (where the difference in dot number between the two squares varied from 1 to 70) with 3 trials each. It collected confidence on an 11-point scale that goes from 1 (certainly wrong) to 11 (certainly correct). However, because the first six confidence ratings were used very infrequently, I combined them into a single rating, thus transforming the 11-point scale into a 6-point scale. On the other hand, Rouault2 used a continuously running staircase that adaptively modulated the difference in dots. It collected confidence on a 6-point scale that goes from 1 (guessing) to 6 (certainly correct), which is equivalent to the modified scale from Rouault1 and thus did not require additional modification. Data collection for both studies was conducted online. Thirty-two participants were excluded from Rouault1 and 13 participants were excluded from Rouault2 (6.43% and 2.62% exclusion rates, respectively).

## Shekhar dataset

The final dataset is named "Shekhar_2021" in the Confidence Database (simplified to "Shekhar" here) and consists of 20 participants each completing 2800 trials. All experimental details can be found in the original publication[27]. Briefly, the task was to determine the orientation (left vs. right) of a Gabor patch presented at fixation. Participants indicated their confidence simultaneously with the perceptual decision using a single mouse click. Confidence was provided on a continuous scale (from 50 to 100) but was binned into six levels as in the original publication. The dataset featured three different difficulty levels (manipulated by changing the contrast of the Gabor patch), which were analyzed separately. No participants were excluded from this dataset.

## Computation of each measure of metacognition

**Previously proposed measures of metacognition.** I computed a total of 17 measures of metacognition and provided Matlab code for their estimation (available at https://osf.io/y5w2d/). I first discuss nine of these measures that have been previously proposed: *AUC2*, *Gamma*, *Phi*, *ΔConf*, *meta-d'*, *M-Ratio*, *M-Diff*, *meta-noise*, and *meta-uncertainty*.

The first four of these measures have the longest history. *AUC2* was first proposed in 1950s[31] and measures the area under the Type 2 ROC function that plots Type 2 hit rate vs. Type 2 false alarm rate. *Gamma* is perhaps the most popular measure in the memory literature and measures are the Goodman–Kruskall Gamma coefficient, which is essentially a rank correlation between trial-by-trial confidence and accuracy[32]. *Phi* is conceptually similar to *Gamma* but measures the Pearson correlation between trial-by-trial confidence and accuracy[33]. Finally, *ΔConf* (my terminology) measures the difference between average confidence on correct trials and the average confidence on error trials. *ΔConf* is perhaps the simplest and most intuitive measure of metacognition but is used very infrequently in the literature.

The next three measures were developed by Maniscalco and Lau[34]. They devised a new approach to measuring metacognitive ability where one can estimate the sensitivity, *meta-d'*, exhibited by the confidence ratings. Because *meta-d'* is expressed in the units of *d'*, Maniscalco and Lau reasoned that *meta-d'* can be normalized by the observed *d'* to obtain either a ratio measure (*M-Ratio*, equal to *meta-d'/d'*) or a difference measure (*M-Diff*, equal to *meta-d'−d'*). These measures are often assumed to be independent of task performance[18] but empirical work on this issue is scarce (though see[41]).

Finally, recent years have seen a concerted effort to build models of metacognition derived from explicit process models of metacognition. Two such measures examined here were developed by Shekhar and Rahnev[27] and Boundy-Singer et al.[35] Shekhar and Rahnev proposed the lognormal meta-noise model, which is an SDT model with the additional assumption of lognormally distributed metacognitive noise that affects the confidence criteria. The lognormal distribution was used because it avoids non-sensical situations where a confidence criterion moves on the other side of the decision criterion. The metacognitive noise parameter ($\sigma_{meta}$, referred to here as *meta-noise*) can be used as a measure of metacognitive ability. The fitting of model to data is rather expensive because it requires the computation of many double integrals that do not have numerical solutions. Consequently, the fitting method from Shekhar and Rahnev[27] takes substantially longer than other measures examined here, making the measure less practical. To address this issue, I make substantial modifications to the original code including many improvements in the efficiency of the algorithm and creating a lookup table so that values of the double integral do not need to be computed anew but can be simply loaded. These improvements reduce the computation of *meta-noise* from minutes to a few seconds, thus making the measure easy to use in practical applications. The measure developed by Boundy-Singer[35]−*meta-uncertainty*−is based on a different process model of metacognition, CASANDRE, that implements the notion that people are uncertain about the uncertainty in their internal representations.

Specifically, it denotes the noise present in the estimation of the sensory noise. The second-order uncertainty parameter, *meta-uncertainty*, represents another possible measure of metacognition. The code for estimating *meta-uncertainty* was provided by Zoe Boundy-Singer.

**New measures of metacognition.** In addition to the already established measures mentioned above, I developed several new measures that conceptually follow the normalization procedure introduced by Maniscalco and Lau[34]. That normalization procedure has previously only been applied to the measure *meta-d'* (to create *M-Ratio* and *M-Diff*), but there is no theoretical reason why a conceptually similar correction cannot be applied to other traditional measures of metacognition. Consequently, here I develop eight new measures where one of the traditional measures of metacognitive ability is turned into either a ratio (*AUC2-Ratio*, *Gamma-Ratio*, *Phi-Ratio*, and *ΔConf-Ratio*) or a difference (*AUC2-Diff*, *Gamma-Diff*, *Phi-Diff*, and *ΔConf-Diff*) measure. The logic is to compute an observed and an expected value for any given measure (e.g., *AUC2*), and then use the expected value to normalize the observed value. First, a measure is computed using the observed data, thus producing what may be called, e.g., $AUC2_{observed}$. Critically, the measure is then computed again using the predictions of SDT given the observed sensitivity (*d'*) and criteria, thus obtaining what may be called, e.g., $AUC2_{expected}$. One can then take either the ratio (e.g., $AUC2_{observed}/AUC2_{expected}$) or the difference (e.g., $AUC2_{observed} − AUC2_{expected}$) between the observed and the SDT-predicted quantities to create the new measures of metacognition.

I computed the SDT expectations in the following way. First, I estimated d' using the formula:

$$d' = z(HR) − z(FAR) \tag{1}$$

where HR is the observed hit rate and FAR is the observed false alarm rate. Then, I estimated the location of all confidence and decision criteria using the formula:

$$c_i = − \frac{z(HR_i) + z(FAR_i)}{2} \tag{2}$$

In the formula above, *i* goes from $−(k − 1)$ to $k − 1$, for confidence ratings collected on an k-point scale. Intuitively, one can think of the confidence ratings $1, 2, \ldots k$ for category 1 being recoded to $−1, − 2, \ldots − k$, such that confidence goes from $−k$ to $k$ and simultaneously indicates the decision (negative confidence values indicating a decision for category 1; positive confidence values indicating a decision for category 2). $HR_i$ and $FAR_i$ are then simply the proportion of times this rescaled confidence is higher or equal to *i* when category 2 and category 1 are presented, respectively.

Once the values of *d'* and $c_i$ are computed, they can be used to generate predicted $HR_i$ and $FAR_i$ values (which would be slightly different from the empirically observed ones). The measures *AUC2*, *Gamma*, *Phi*, and *ΔConf* can then be straightforwardly computed based on the predicted $HR_i$ and $FAR_i$ values, thus enabling the computation of the new ratio and difference measures.

## Assessing validity and precision

Any measure of metacognition should be valid and precise[19,22,48]. However, there is no established method to assess either validity or precision of measures of metacognition. Here I developed a method to jointly assess validity and precision. The underlying idea is to artificially alter confidence to be less in line with accuracy and then assess how measures of metacognition change.

Specifically, the method corrupts confidence by decreasing confidence ratings for correct trials and increasing them for incorrect trials. For a given set of trials, the method loops over the trials starting from the first and (1) if the trial has a correct response and confidence

higher than 1, then it decreases the confidence on that trial by 1 point, and (2) if the trial has an incorrect response and confidence lower than maximum (that is $k$ on an $k$-point scale), then it increases the confidence on that trial by 1 point. If neither of these conditions apply, the trial is skipped. The method then continues to corrupt subsequent trials in the same manner until a pre-set proportion of corrupted trials is achieved. Then, all measures of metacognition are computed based on the corrupted confidence ratings. A given dataset is first split into $n$ bins of a given trial number, and the procedure above is performed separately for each bin. Finally, to compute a measure of precision that can be compared across different measures of metacognition, I use the following formula:

$$precision = \frac{1}{n}\sum_{i=1}^{n}\frac{measureOrig_i - measureCorrupted_i}{SD} \qquad (3)$$

where $measureOrig_i$ and $measureCorrupted_i$ are the values of a specific measure computed on the original (uncorrupted) and corrupted confidence ratings, respectively, $n$ is the number of bins analyzed, and $SD$ is the standard deviation of all $measureOrig_i$ for $i = 1, 2, \ldots, n$. Positive values of the variable $precision$ indicate valid measures of metacognition and higher values indicate more precise measures (e.g., measures more sensitive to corruption in confidence compared to background fluctuations).

I computed the precision of all 17 measures of metacognition for two datasets from the Confidence Database: Maniscalco (1 day; 1000 trials per participant) and Haddara (6 days; 3000 trials per participant). I separately examined the results of altering 2, 4, and 6% of all trials and computed metacognitive scores based on bins of 50, 100, 200, and 400 trials. I split the Maniscalco dataset into 20 bins of 50 trials, 10 bins of 100 trials, five bins of 200 trials, and two bins of 400 trials (by taking into consideration only the first 800 trials in this last case). I split the 500 trials from each of the six days in the Haddara dataset into 10 bins of 50 trials, five bins of 100 trials, two bins of 200 trials, and one bin of 400 trials (by taking into consideration only the first 400 trials for the 200- and 400-trial bins). Across the 6 days, this process resulted in 60 bins of 50 trials, 30 bins of 100 trials, 12 bins of 200 trials, and six bins of 400 trials.

## Assessing dependence on task performance
To assess how task performance affects measures of metacognition, I examined whether each measure of metacognition changed across different difficulty levels in the same experiment. Specifically, I tested whether each of the 17 measures of metacognition increases or decreases for more difficult conditions. This process requires datasets with (1) several difficulty conditions and (2) a large number of trials. Consequently, I selected datasets from the Confidence Database that meet these two criteria but do not include any other manipulations. This resulted in the selection of three datasets: Shekhar (3 difficulty levels, 20 participants, 2800 trials/participant, 56,000 total trials), Rouault1 (70 difficulty levels, 466 participants, 210 trials/participant, 97,860 total trials), and Rouault2 (many difficulty levels, 484 participants, 210 trials/participant, 101,640 total trials). Because the two Rouault datasets included very few trials from each difficulty level, I instead used a median split to classify them in easy vs. difficult. To perform statistical analyses and compute Cohen's $d$, I conducted t-tests comparing the lowest and highest difficulty levels in each dataset. To avoid outlier values, for each difficulty level and each measure of metacognition, I excluded any values that deviated by more than 3*SD from the mean of that difficulty level. Finally, as a reference, I performed all the above analyses on the measures $d'$, $c$, and average confidence.

## Assessing dependence on metacognitive bias
To assess how metacognitive bias affects measures of metacognition, I applied the method developed by Xue et al.[28]. In this method, confidence ratings are recoded in two different ways as to artificially induce metacognitive bias towards lower or higher confidence ratings. Specifically, an n-point scale is transformed into an $(n-1)$-point scale in two ways. In the first recoding, the ratings from 2 to $n$ are all decreased by one. In the second recoding, only the rating of $n$ is decreased by one. When compared to each other, the first method results in a bias towards lower confidence compared to the second method (see mean confidence values in the bottom right of Fig. 3a). A measure of metacognition can then be computed for the newly obtained confidence ratings. Comparing the obtained values for the two recodings allows the assessment of whether each measure of metacognition is independent of metacognitive bias.

This process would ideally be applied to datasets with (1) a single experimental condition and (2) a large number of trials. Consequently, I selected the same two datasets used to quantify precision: Haddara (3000 trials per participant) and Maniscalco (1000 trials per participant). In addition, I also used the Shekhar dataset (3 difficulty levels, 2800 trials per participant) but analyzed each difficulty level in isolation and then averaged the results across the three difficulty levels. The values of each measure of metacognition for the two recodings were compared using a paired t-test.

## Assessing dependence on response bias
To assess how response bias affects measures of metacognition, I compared the values of each measure of metacognition in conditions that differed in their decision criterion. To do so, I analyzed the Locke dataset—the only dataset in the Confidence Database where the response criterion is experimentally manipulated. I computed each measure of metacognition for each of the seven conditions in that dataset and conducted repeated measures ANOVAs to examine whether each measure of metacognition varied with the condition. In addition, to estimate an effect size for the relationship between response bias and each measure of metacognition, I computed the correlation between the estimated metacognitive ability and the absolute value of the response bias (i.e., $|c|$).

## Assessing split-half reliability
To assess split-half reliability, I examined the correlation between the values obtained for different measures of metacognition on odd vs. even trials[41]. As with assessing precision, I estimated split-half correlations for different sample sizes, so researchers can make informed decisions about the sample sizes needed in future studies. Specifically, I used bin sizes of 50, 100, 200, and 400 trials. Note that a bin size of $k$ here means that $2k$ trials were examined with both the odd and even trials having a sample size of $k$. These computations are best performed using datasets with (1) a single condition, and (2) a large number of trials per participant. Consequently, I selected the same three datasets used to examine the dependence of measures of metacognition on metacognitive bias: Haddara (3000 trials per participant), Maniscalco (1000 trials per participant), and Shekhar (3 difficulty levels, 2800 trials per participant). As before, I analyzed each difficulty level in the Shekhar dataset in isolation and then averaged the results across the three difficulty levels. For a bin size of $k$, the computations were performed on as many as possible non-overlapping bins of $2k$ trials. The obtained r-values were then z-transformed, averaged, and the resulting average $z$ value was transformed back to an r-value for reporting and plotting purposes.

## Assessing test-retest reliability
To assess test-retest reliability, I examined the intraclass correlation (ICC) coefficients between the values obtained for different

measures of metacognition on different days. I report the two-way absolute consistency ICC, named "A-1"[62] computed using the code provided by Salarian (https://www.mathworks.com/matlabcentral/fileexchange/22099-intraclass-correlationcoefficient-icc). For ease of comparison with the results by Guggenmos[41], in addition to ICC, I also computed the Pearson correlation. As with split-half reliability, I estimated test-retest reliability for sample sizes of 50, 100, 200, and 400 trials. Because test-retest computations require data from multiple days and a large number of trials per participant per day, I selected the Haddara dataset as it is the only dataset in the Confidence Database to meet these criteria. I computed test-retest correlations between all pairs of days for as many as possible non-overlapping bins. Note that, unlike for split-half analyses, analyses of bin size of $k$ involved the selection of $k$ trials from each day. As with the split-half analyses, the obtained correlation coefficients were then z-transformed, averaged, and the resulting average $z$ value was transformed back to a correlation coefficient (ICC or $r$-value) for reporting and plotting purposes.

### Statistical analyses and reporting

All conclusions in the paper are based on effect sizes (Cohen's $d$, $r$, and ICC values). However, for completeness, I sometimes refer to the results of null-hypothesis statistical tests. As is standard practice, in cases where I report the results of multiple tests together, I only include the $p$-values. All remaining information, such as test statistics and degrees of freedom, can be obtained from the provided analysis codes. All $p$-values are based on two-tailed statistical tests. Analyses were performed using MATLAB 2024a (MathWorks).

### Reporting summary

Further information on research design is available in the Nature Portfolio Reporting Summary linked to this article.

## Data availability

Raw data for all six experiments were obtained from the Confidence Database (https://osf.io/s46pr). The data come from the following publications: Haddara and Rahnev[43], Locke et al.[44], Maniscalco et al.[45], Rouault et al.[46], and Shekhar and Rahnev[27]. Processed data files are available at https://doi.org/10.17605/osf.io/y5w2d.

## Code availability

Code for computing all 17 measures of metacognition, as well as data and analysis code for reproducing all statistical results and plotting all figures are available at https://doi.org/10.17605/osf.io/y5w2d.

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

## Acknowledgements
This work was supported by the National Institute of Health (award R01MH119189) and the Office of Naval Research (award N00014-20-1-2622). The author thanks Zoe Boundy-Singer for sharing scripts for estimating meta-uncertainty.

## Author contributions
D.R. is the sole author and performed all work related to this paper.

## Competing interests
The author declares no competing interests.
