## [Transparent Peer Review file · Nature Communications]

A comprehensive assessment of current methods for measuring metacognition

Corresponding Author: Dr Dobromir Rahnev

Version 0:

Reviewer comments:

Reviewer #1

(Remarks to the Author)

The author evaluates an extensive set of methods that can be used to estimate a person's "metacognitive ability". Many of the methods have been suggested in previous research, but some are newly created for the purposes of comparison. The methods are assessed in terms of their sensitivity to factors that should affect a good measure of metacognitive ability, their insensitivity to nuisance factors, and their reliability in split-half and test-retest comparisons. The results reproduce some known effects, such as the sensitivity of simple measures of metacognitive accuracy to task difficulty and the low test-retest reliability of some measures, and include some new tests such as sensitivity to response bias. The results identify important variability across the methods in terms of their sensitivity and stability. The author concludes with practical recommendations using the measures according to the question of interest.

The manuscript has many strengths. The research is timely given the current high levels of interest in confidence and metacognition. The approach is thorough in terms of the number and range of methods considered. The presentation is clear. The evaluation is balanced and fair. The manuscript also has some important weaknesses. For me, the main one is the approach to measuring a method's sensitivity to nuisance factors, which depends on comparisons of significance across different statistical tests. As well as this, the target concept of "metacognitive ability" could be more clearly defined, and the relationship between bias and ability explored further and clarified. These and other more minor points are detailed below.

Major comments

1) I wasn't convinced by the analyses of methods' sensitivity to nuisance factors. The general approach is to assess which methods show significant effects of manipulations of those nuisance variables. However, comparing statistical significance is invalid in general (e.g. Nieuwenhuis et al., 2011, Nature Neuroscience), and many of the problems of this approach are apparent here. For example, based on the results summarized in Figure 2 the author concludes that AUC2-ratio is more sensitive to task performance than other ratio measures. But all 5 ratio measures show a very similar pattern of dependence (e.g. the U-shaped dependence in the Adler dataset) and it would be very surprising if direct comparison across methods (however that might be done) would show a greater dependence of AUC2-ratio than the other methods. By eye, it looks like the difference in statistical significance depends as much on higher variability (wider error bars) in the other measures as it does on differences in effect magnitude. This problem, of statistical tests being sensitive to noise as well as effect size, runs through the sections on nuisance factors, not only complicating contrasts across tests but also undermining interpretations of the data presented. For example, the non-significant dependence of meta-noise on metacognitive bias and response bias seems to reflect more the noise in this measure than its stability across conditions. In Figure 3, the meta-noise estimates in the Maniscalco dataset differ by a factor of ~2 for low vs high bias. In Figure 4, the estimates for the first two plotted conditions are ~0.5 and ~1.5, implying a large numerical change in metacognitive ability. This variability implies that numerical estimates of ability using this measure are very unreliable, but the lack of significance is interpreted as a positive for the method. Extending this point, some analyses are undermined further by the particular statistical test chosen. In particular the analysis of task performance only assesses linear effects of difficulty, but clear quadratic trends are clear in the Adler data. Although the standard worry is that metacognitive ability will be underestimated when the task is hard and overestimated when it is easy, it is still problematic if metacognitive ability is systematically under/overestimated when difficulty is at either extreme, the pattern seen in Figure 2 for all methods showing a non-significant linear effect in the Adler dataset. To present a convincing case, the analyses would need to consider more direct contrasts across measures and otherwise deal with the issue of variability of estimates.

2) The main concept of “metacognitive ability” could be clarified. The core idea seems to be that there is a stable trait in individuals that will endure across days at least (as implied by the test-retest reliability analysis), but beyond this I wasn’t sure how the reader should conceive of metacognitive ability. Is it also expected to be stable for an individual across tasks, modalities, cognitive domains, etc.? If so, what is the hypothesized basis of the trait? For example, it might depend on the integrity of a particular neural circuit or be derived from other personality traits such as conscientiousness. Providing evidence on these questions is of course well beyond the scope of this study, but an early statement defining metacognitive ability (at least the author’s working model of it) would help readers who aren’t already immersed in this literature. Somewhat related, I was initially confused by the treatment of task difficulty effects as a “nuisance variable” given its apparently dominating effect on metacognitive performance. In many practical settings, if most of the variance in observed metacognitive performance is explained by a person’s task performance rather than their trait metacognitive ability, then isn’t the observable metacognitive performance (rather some latent ‘ability’) what matters? Eventually, I understood that task performance is a nuisance variable in relation to estimating a hypothesized stable trait ability, but I would’ve understood that perspective earlier with a clearer statement of what this trait might be.

3) The manuscript would benefit from some clarification about the relationship between metacognitive bias and ability. Currently they are treated as separate, with bias characterized as a nuisance variable, but I am not sure they can be fully distinct. For example, if a person is truly extremely biased, for example in terms of being consistently highly confident, then aren’t they necessarily also low in metacognitive ability? Beyond this theoretical perspective, there are also practical aspects of this question. The method used to test metacognitive biases, described on p.53, involves recoding confidence values, for example, by recoding all values from 2 to n-1 scale as having the value n-1 (i.e. collapsing values 1 and 2 together). This “bias” should necessarily reduce metacognitive ability to the degree that there is meaningful difference between ratings of 1 and 2, yet the analysis seems to be based on the idea that a good measure will be insensitive to this collapse. Clarification of this would be helpful. Then in Figure 3 it would be useful to see the estimated metacognitive ability values in the original data (prior to recoding). I assume they are all higher. Where “The results demonstrate that meta-d’, AUC2, Phi, and Δ Conf tend to increase with higher average confidence”, is the reason that the tasks were hard, so participants were mostly using low confidence values, so that collapsing values at this end of the scale had a bigger impact?

Minor comments

p.3. Metacognition is more than confidence in decisions. Metacognition has been widely studied in terms of learning, memory, problem solving, everyday abilities, among others.

Is the analysis extendable to these?

p.4. Is really true that validity has been ignored? Most/all measures in the literature assess the degree to which metacognitive evaluations match objective reality. Isn’t that a face valid approach?

p.11. Do the analyses listed in Table 4 correspond to the ones originally conducted by the study authors, or are they the ones done here?

p.17. The pairwise comparisons should be post hoc tests, because they are run based on inspection of the data. Also here, can the author provide any intuition for why meta-uncertainty does so much worse?

p.20. “The reason why AUC2-Ratio does not perform well is that AUC2, unlike the other four measures here, has a lower boundary of 0.5 rather than 0.” Please explain why this feature leads to poorer performance, because it is not intuitively clear.

p.23. “Given that meta-noise corrected in the opposite direction of the other five measures, it may be advisable for any result to be reproduced both meta-noise and at least one more measure from the list above”. It would greatly strengthen this recommendation if it was explained why meta-noise behaved in this way, so that the reader is reassured that the opposite direction is a feature not a bug of this method.

p.46. For readers unfamiliar with the original work, it would be helpful to give an intuition for why a lognormal distribution is used in the meta-noise model, and what it implies about metacognition. Similarly for the meta-uncertainty model, it would be useful to give an indication of how uncertainty about uncertainty affects metacognitive reports (and/or how this is estimated).

p.51. “If multiple difficulty conditions were present in a given dataset, I performed a linear regression on the values of each measure and then ran a one-sample t-test against 0 on the resulting slopes.” Please clarify the description of the analysis. Is a separate regression fit per participant and then second-order statistics performed on the resulting regression betas?

p.53. “In the second recoding, only the rating of n is decreased by one, resulting in a bias towards high confidence.” Wouldn’t this manipulation decrease confidence? Is the idea that this is equivalent to collapsing at the high end of the scale?

Typos and language

Check and correct formatting of in-text citations such as “To address this issue, (Maniscalco & Lau, 2012) developed...”.

p.11. “Dependance” in Table 4 but “dependence” elsewhere.

p.16, Figure 1. The panels aren’t labeled in a way that matches the figure caption. What are the horizontal lines in the plots of “Decrease in SD units”?

p.31. Figure 6 shouldn’t be a line graph because neighboring points aren’t theoretically linked any more than non-neighboring ones.

p.46. “((Kornell et al., 2007))”

Reviewer #2

(Remarks to the Author)

Review of manuscript NCOMMS-23-32257 titled “Measuring metacognition: A comprehensive assessment of current

methods”.

In this manuscript the author compares several methods for measuring metacognitive sensitivity across several public data sets. The manuscript is well written and will be useful for researchers studying metacognition. I present below my recommendations for improving the manuscript in what is my standard review structure.

1. Do the conclusions follow from the data?

The author provides many useful tests of current measures of metacognition, with useful recommendations for moving forward.

2. Is the analysis conducted appropriately?

When testing for reliability in test-retest or split-half data, an absolute agreement intraclass correlation coefficient (ICC; Shrout & Fleiss, 1979) is more appropriate than the Pearson correlation coefficient. The Pearson correlation coefficient tests $y = mx + b$, how well x predict y , allowing x and y to have different means (differing by b). The absolute agreement ICC, as the name suggests, tests $y = x$, how well x predicts y assuming they should be the same. The author may wish to keep the Pearson's r correlation coefficient for comparison with Guggenmos 2021 (or use the ICC and the normalised mean absolute error, the other reliability measure used in Guggenmos, 2021). See Matheson (2019) for more detailed explanation of ICC in test-retest contexts.

A Fisher transform should be applied to correlation coefficients prior to averaging. Correlation coefficients are bounded and therefore not normally distributed. Toward the bounds the distributions become skewed and therefore the mean is inappropriate. The Fisher transform gives a normally distributed scalar of correlation coefficients (the inverse can be applied to the average).

The analysis referring to 'task-performance' actually only indirectly applies to performance via externally manipulated difficulty. The author should rephrase to refer to task-difficulty throughout, or perform the analysis on task-performance (d'). An analysis of task difficulty would make the comparisons across data-sets better, and be more direct (for example, difficulty levels 5 and 6 of the Adler data don't seem to differ in terms of task performance). The author may also note that some readers may be confused by the x-axis of Figure 2 in that increasing numerical label of difficulty corresponds to increasing performance, presumably because it is actually decreased difficulty.

I found myself wondering if the ratio measures are comparable? They should be in similar enough units for a Pearson correlation. I think other readers would be interested if the author is willing to add this comparison.

Contrary to the reporting summary, the test statistics, confidence intervals and degrees of freedom are not disclosed for the tests of dependence on 'task performance', dependence on metacognitive bias, and dependence on response bias (could give the range of values in cases where many tests were performed).

3. Do the introduction and methods give sufficient information for understanding the experiment?

The introduction is well structured and clear (with the exception of the minor issues below). The methods also gives sufficient detail.

4. Does the discussion sufficiently outline the findings?

The discussion is well rounded and also very clear.

Minor:

The definition of metacognition (first sentence of introduction) does not match the reference (who define metacognition as knowing about knowing – the book is largely about metamemory).

“Several papers have used simulations to investigate some of the properties of measures of metacognition (Barrett et al., 2013; Guggenmos, 2021), but this approach is potentially problematic because it is a priori unknown how well the process models used to simulate data reflect empirical reality.” This sentence implies that the Guggenmos paper did not examine data also. The author may wish to consider painting it in a more positive light given many of the comparisons made there are duplicated in the current manuscript.

“One paper (Azzopardi & Evans, 2007) examined the properties of a measure, Type-2 d' , which was subsequently shown to be based on faulty assumptions (Galvin et al., 2003)...” 2003 is not subsequent to 2007. Azzopardi and Evans actually investigated Kunimoto's a' , showing it to be corrupted by response bias. This issue of bias was popular at the time; Persuade and colleagues (2007) published a wagering method (similar to the Kunimoto 2001 method), which was rapidly debunked by Clifford et al. (2008). The Galvin paper is about type-2 ROCs, listing many caveats about confidence ratings that make ROC-like analyses (including meta- d') based on potentially false assumptions. Azzopardi and Evans discuss this in their paper. This is to say, there are interesting things about the references in the sentence, but the sentence is incorrect.

“The one exception was meta-uncertainty, which had substantially lower average precision score in both the Haddara (meta-uncertainty: 0.37; average of other measures: 0.67) and the Maniscalco datasets (meta-uncertainty: 0.30; average of other measures: 0.53).” Boundy-Singer et al., 2022 themselves note issues of parameter recovery for small numbers of trials (100) and recommend at least 700 trials for this model. The author may wish to note this (none of these tests were performed according to the recommendations of the authors for the meta-uncertainty model).

“within-subject studies of metacognition do not require high reliability – a measure that inherently depends on a large spread of scores across subjects” How do within-subjects studies require spread between subjects?

“I computed a total of 17 measures of metacognition and provide Matlab code for their estimation” It would be nice to have the link with this sentence (in addition to the end of the document).

References in review not in manuscript:

Shrout & Fleiss (1979) Intraclass correlations: uses in assessing rater reliability. *Psychological Bulletin*. 1979;86(2):420–428. doi: 10.1037/0033-2909.86.2.420.

Matheson GJ. We need to talk about reliability: making better use of test-retest studies for study design and interpretation. *PeerJ*. 2019 May 24;7:e6918. doi: 10.7717/peerj.6918. PMID: 31179173; PMCID: PMC6536112.

Reviewer #3

(Remarks to the Author)

Comments on Rahnev et al., “Measuring metacognition: A comprehensive assessment of current methods.” – *Nature Communications*

Overall comments:

This submission addresses an important topic in contemporary cognitive science – metacognition, and how it should be measured. The authors assess most of the currently available metrics of metacognitive sensitivity, using a sensible set of measures. Overall, they find that most of these existing measures are reasonably interchangeable on these metrics. Whether this leaves the reader feeling comforted or despondent may depend on whether they take a glass half full or brimming perspective of the world, and of this area of research.

Overall, I think this submission is a welcome addition, building considerably on the scope of other recent, similar publications. It will be a welcome addition to the public record.

My only reasonably substantive points relate to terminology, and to requests that the authors consider adding to their discussion to address conceptual issues.

Specific Points

1) High precision, or High sensitivity?

The authors refer to High precision as a desirable quality that they are measuring, but to my mind their description and use of this term describes the sensitivity of metacognitive sensitivity tests, not their precision.

To measure ‘precision’ / sensitivity, the authors add a counter measure to ratings of confidence in data, and quantify how far this diminishes evidence of metacognitive sensitivity relative to unperturbed datasets. This speaks to the sensitivity of the measures (i.e. if they are sensitive to metacognitive sensitivity at all), but not to the precision of these measures (i.e. are they just measuring metacognitive sensitivity, or are they also measuring other related constructs).

2) Why should measures of metacognitive sensitivity be independent of task difficulty?

The authors assert that a good measure of metacognitive sensitivity should be independent of task difficulty. I question this assertion and cannot see any underlying logic to it.

Cognitive load is a standard manipulation, used to undermine performance in cognitive tasks. Moreover, there does not need to be a great overlap between cognitive operations for a cognitive load (from concurrently performing a difficult cognitive operation) to undermine performance on another task. Similarly, I can see no reason why the demand to perform a difficult cognitive operation could not undermine a person’s ability to attend to the nuances of their feelings of decisional confidence – even if insight into these is somewhat independent of decisions themselves.

Another way to express this concern is that our feelings of confidence could obey a Weber fraction, in which case it would be

harder (in absolute terms) to distinguish between feelings of confidence that signal large magnitudes of uncertainty, relative to when we distinguish between feelings of confidence that signal small amounts of uncertainty.

I would encourage the authors to consider these points, and to address them in a revised manuscript.

3) Of course, all the measures have similar outcomes. They are all very similar.

All the measures the authors consider are variants on signal-detection-theory (SDT) based instruments. The conceptual differences between the measures are very slight, and so it seems reasonable that the different outcomes of the analyses here are similarly slight.

There are increasingly sophisticated process models, some of which the authors themselves have developed, that are more nuanced. Such approaches have found benefit to the presumption of different shaped noise distributions, and conceptually they could offer some insight into the temporal evolution of a decision process. Here the authors want to unfold a standard set of tests, so that each measure can be equally assessed. But this conceptually levelling of the playing field by the development of some generic tests could create a false impression of equivalence. By reducing all accounts to a generic level that can equally be assessed, the authors may have assured that no measure can substantially outperform its SDT basis.

I think the authors should discuss this issue in a revised manuscript.

Version 1:

Reviewer comments:

Reviewer #1

(Remarks to the Author)

The author has produced a very thorough and thoughtful response to the first set of reviews – thank you for being so responsive my original comments. I am convinced by the additions and edits and am now happy to recommend publication. I identified only a few minor issues to address:

p.24. “meta-uncertainty increased for easier conditions in all three datasets”. This doesn’t seem to be true in the Shekhar dataset, where the value is lowest in the condition with highest d-prime.

p.45. “A possible disadvantage of both measures is that they can only take positive values and therefore cannot be used for situations where metacognition may contain more information than the decision itself”. It may be helpful to expand this statement, e.g. by giving an intuition or indication what it could mean for metacognition to “contain more information” than a decision, and under what circumstances this can occur.

p.48. “Finally, researchers should interpret findings of M-Ratio being larger (or smaller) than 1 with caution”. I wasn’t sure which aspects of the presented results are the basis for this recommendation. None of the figures include M-Ratio values above 1. Maybe this paragraph is a legacy from analyses in the previous version and needs to be cut in this revision?

(Remarks on code availability)

Reviewer #2

(Remarks to the Author)

Review of manuscript NCOMMS-23-32257 titled “Measuring metacognition: A comprehensive assessment of current methods”.

The author has responded constructively to my previous comments and the comments of the other reviewers, which has improved the manuscript.

Only one of my previous comments was not quite addressed: “When testing for reliability in test-retest or split-half data, an absolute agreement intraclass correlation coefficient (ICC; Shrout & Fleiss, 1979) is more appropriate than the Pearson correlation coefficient... The absolute agreement ICC, as the name suggests, tests $y = x$, how well x predicts y assuming they should be the same.”

An ICC was added, but it was a consistency ICC, not an absolute agreement ICC. The consistency ICC also discounts changes in the mean, so would suggest high test-retest reliability if most subjects showed significantly decreased (or increased) metacognitive sensitivity from test 1 to test 2 but by similar amounts. Normally I would consider test-retest reliability to be high if subjects show the same scores from test 1 to test 2, this is what the absolute agreement ICC tests for. In the package the author used, this model is ‘A-1’. The consistency ICC gives very similar results to the Pearson correlation because it is a very similar test.

(Remarks on code availability)

Reviewer #4

(Remarks to the Author)

I greatly enjoyed reading this article, which makes an important contribution to the literature by comparing several metrics aimed at best quantifying metacognitive performance. It will be a handy resource for the community, which is often at a loss when choosing a particular metric.

The author has already done a great deal of work to explain the complex concepts discussed in the article. He has also gone to great lengths to respond to reviewers. The result is excellent, and I have only minor comments to make, which you'll find below.

General Comments:

1. Although mentioned and discussed satisfactorily in the Discussion section (Limitations), the fact that most datasets and measures are designed for two-choice tasks seems relevant. I believe it should also at least be mentioned when introducing the datasets on page 16.
2. Page 9 when introducing the second nuisance variable, you have the sentence: "The logic here is similar to the logic in SDT, where the measure of performance is designed to be mathematically independent of the measure of response bias." In my opinion, it makes the text confusing to have response bias used here before it is introduced, even more so when talking about metacognitive bias for the first time. Consider switching the order of introducing the variables metacognitive bias and response bias.
3. Figure 7 (Page 36), I find this Figure hard to read due to the x-axis labels. Perhaps it would be best to use the 7 numerical categories introduced in the text on page 35 (1 to 7).
4. Page 37, Figure 7's caption says "While condition strongly modulated response bias..." This is expected, right? If I understand correctly, this result is like a sanity check to compare response bias in the experimental setup (condition) and response bias as a metric (criterion, c), but please clarify if I am understanding something wrongly. Having said this, I have a concern with the mix between the words response bias (criterion, c) and the response bias associated with condition in the experimental setup. I believe it would be a good idea to be careful with the naming convention and avoid ambiguities, given that it is a dense and complex text. Would it be better to always call criterion (c), criterion, and just use response bias to talk about the condition in the experimental setup? In any case, I think clarification would help. For example, the caption in Figure 7 would then change to "While response bias strongly modulated criterion, c, (...)".
5. The author newly incorporated a correlation test (ICC) in addition to the Pearson correlation for test-retest reliability. Figure 10 seems to show the same graph for both measurements, could this be a mistake? Otherwise, if the graphs are almost identical, I find the Pearson correlation figure to be unnecessary and could be removed from the paper.

Typos and minor

1. Page 6, Table 1 is either missing a full-stop (period) in the second row.
3. I see an inconsistency in the use of decimal values within the text, for example, In page 44, the across subject correlation is reported as "average $r = .49, .55, (...)$ ", but the inter-measures correlation as " $r = 0.60, 0.63, (...)$ ". As said before, the complexity of the topic requires attention and good understanding, I believe it would be easier to read if this was consistent, personally prefer 0.xx format.
4. Figure 12m Page 48. There is an extra comma, where it says "The figure lists, the values (...)" it should say "The figure lists the values (...)".
5. I would recommend using the symbol for d' consistently throughout the text. Sometimes it is d' and sometimes it is $d'.$,

(Remarks on code availability)

Reviewer #5

(Remarks to the Author)

(Remarks on code availability)

Version 2:

Reviewer comments:

Reviewer #2

(Remarks to the Author)

The author has addressed all my comments, I congratulate them on the manuscript.

(Remarks on code availability)

Thank you for sending the paper for review and securing reviewers who are clearly experts in the field. I found the comments of the reviewers very helpful and constructive. Based on these comments, I have conducted several new analyses, prepared a number of new figures (5 in the main text and 3 in the Supplementary), and enhanced the presentation of the paper. I believe that this process has substantially improved the manuscript. I'm including a point-by-point response to all comments below.

Sincerely,
Doby Rahnev

Reviewer 1

The author evaluates an extensive set of methods that can be used to estimate a person's "metacognitive ability". Many of the methods have been suggested in previous research, but some are newly created for the purposes of comparison. The methods are assessed in terms of their sensitivity to factors that should affect a good measure of metacognitive ability, their insensitivity to nuisance factors, and their reliability in split-half and test-retest comparisons. The results reproduce some known effects, such as the sensitivity of simple measures of metacognitive accuracy to task difficulty and the low test-retest reliability of some measures, and include some new tests such as sensitivity to response bias. The results identify important variability across the methods in terms of their sensitivity and stability. The author concludes with practical recommendations using the measures according to the question of interest.

The manuscript has many strengths. The research is timely given the current high levels of interest in confidence and metacognition. The approach is thorough in terms of the number and range of methods considered. The presentation is clear. The evaluation is balanced and fair. The manuscript also has some important weaknesses. For me, the main one is the approach to measuring a method's sensitivity to nuisance factors, which depends on comparisons of significance across different statistical tests. As well as this, the target concept of "metacognitive ability" could be more clearly defined, and the relationship between bias and ability explored further and clarified. These and other more minor points are detailed below.

I thank the reviewer for appreciating the strengths of the manuscript and for bringing up important areas for improvement. I have now moved away from comparing significance across different statistical tests, clarified the concept of "metacognitive ability," explored further the relationship between bias and ability, and addressed all remaining points. I expand on each of these points in my responses below.

Major comments

1) I wasn't convinced by the analyses of methods' sensitivity to nuisance factors. The general approach is to assess which methods show significant effects of manipulations of those nuisance variables. However, comparing statistical significance is invalid in general (e.g. Nieuwenhuis et al., 2011, Nature Neuroscience), and many of the problems of this approach are apparent here. For example, based on the results summarized in Figure 2 the author concludes that AUC2-ratio is more sensitive to task performance than other ratio measures. But all 5 ratio measures show a very similar pattern of dependence (e.g. the U-shaped dependence in the Adler dataset) and it would be very surprising if direct comparison across methods (however that might be done) would show a greater dependence of AUC2-ratio than the other methods. By eye, it looks like the difference in statistical significance depends as much on higher variability (wider error bars) in the other measures as it does on differences in effect magnitude. This problem, of statistical tests being sensitive to noise as well as effect size, runs through the sections on nuisance factors, not only complicating contrasts across tests but also undermining interpretations of the data presented. For example, the non-significant dependence of meta-noise on metacognitive bias and response bias seems to reflect more the noise in this measure than its stability across conditions. In Figure 3, the meta-noise estimates in the Maniscalco dataset differ by a factor of ~ 2 for low vs high bias. In Figure 4, the estimates for the first two plotted conditions are ~ 0.5 and ~ 1.5 , implying a large numerical change in metacognitive ability. This variability implies that numerical estimates of ability using this measure are very unreliable, but the lack of significance is interpreted as a positive for the method. Extending this point, some analyses are undermined further by the particular statistical test chosen. In particular the analysis of task performance only assesses linear effects of difficulty, but clear quadratic trends are clear in the Adler data. Although the standard worry is that metacognitive ability will be underestimated when the task is hard and overestimated when it is easy, it is still problematic if metacognitive ability is systematically under/overestimated when difficulty is at either extreme, the pattern seen in Figure 2 for all methods showing a non-significant linear effect in the Adler dataset. To present a convincing case, the analyses would need to consider more direct contrasts across measures and otherwise deal with the issue of variability of estimates.

I thank the reviewer for pointing out this important point. I agree that the previous version of the manuscript relied too much on statistical significance to make conclusions. I was uncomfortable with that myself, but hadn't spent the time to think through possible alternatives. At the urging of the reviewer, I have now done so. While I still sometimes report statistical significance for completeness, the conclusions are now drawn based exclusively on the observed effect sizes (Cohen's d , r -values, or ICC values). In fact, I created 3 new figures – one for each nuisance factor – to show the comparisons in the context of Cohen's d or r -values.

One possible limitation of this approach – as noted by the reviewer – is that noisier metrics will have an inherent advantage in these analyses because their noisiness increases the denominator in the formula for Cohen's d . What can be done about this? Unfortunately, I can't find a solution and I suspect no solution better than reporting Cohen's d exists when it comes to meaningfully comparing the measures to each other. The obvious possible solution is to

report raw values of each metric (e.g., a manipulation changes meta- d' by 0.4 and AUC2 by 0.1). However, because the different measures live on vastly different scales, such information is informative for each measure in isolation but cannot be used to draw meaningful comparisons between measures. Alternatively, one can look at ratios, but that is problematic too. For example, consider a situation where a metric is 0.01 in one condition and 0.05 in the other condition. One may reasonably say that there's little difference between the conditions, but the ratio will be a whopping 5. Alternatively, AUC2 goes from 0.5 to 1, so even a very large difference (say, between 0.7 and 0.8) would only result in a small ratio difference. Again, the measures are simply not meaningfully comparable if one is to consider the ratio in isolation.

But one can respond: Isn't Cohen's d also useless given the apparent difference in noisiness in the measures? Here I would contend that the earlier analyses on precision ensure that analyzing Cohen's d is meaningful. What the precision analyses showed is that 16 out of the 17 measures have very similar precision levels. In other words, 16 of the 17 measures are almost equally noisy. Of course, in any specific analysis or test, it may appear that one measure is noisier than others, but given the results on precision, these differences must be chance occurrences rather than systematic differences between the measures.

The one exception to this logic is meta-uncertainty, which has substantially lower precision and would thus be favored in the analyses of Cohen's d (because of its larger denominator). While this is not ideal, it is much easier to qualify and put into context the results for a single measure if all remaining measures are already compared fairly to each other. I also note that meta-uncertainty has never been used outside of the original paper that introduced it, so it seems unreasonable to eschew Cohen's d only because it leads to interpretational issues with that single measure.

Finally, the reviewer noted the weirdness of Adler's dataset. I have now examined that dataset in more detail and realized that many subjects show very unstable values, presumably because of the complexity of the original design. Given that the dataset is also somewhat small ($N=19$), I have decided to remove it from the paper. This still leaves 3 datasets for the analyses of task difficulty and they paint a very consistent picture, meaning that dropping the Adler dataset can be done with little loss. The other dataset with multiple difficulty values (Shekhar) never shows quadratic effects, and correspondingly I don't include quadratic fits to it.

I have now thoroughly edited the sections on nuisance factors to move away from simple significance reporting, included new graphs with comparisons of Cohen's d or r -values, and have included detailed discussion of the issues raised here. I believe that the new section does a much better job at meaningfully comparing the different measures. I paste the relevant edited sections (new figures, new/edited parts of the Results, and new Discussion part) below:

New figures

Figure 4. Effect sizes for dependence on task performance. Effect sizes (Cohen’s d) is plotted for each metric and dataset. As can be seen in the figure, non-normalized traditional measures (i.e., $meta-d$, $AUC2$, $Gamma$, Phi , and $\Delta Conf$) show strong positive relationship with task performance. Corrections with the Ratio and Diff methods reverse this relationship, with the Ratio correction being clearly superior. The model-based metrics $meta-noise$ and $meta-uncertainty$ perform well too, with $meta-uncertainty$ showing particularly low effect sizes.

Figure 6. Effect sizes for dependence on metacognitive bias. Effect sizes (Cohen’s d) is plotted for each metric and dataset. As can be seen in the figure, all metrics except for $Gamma$ and $meta-noise$ have a mostly positive relationship with metacognitive bias (i.e., higher confidence leads to higher estimates of metacognition). The smallest absolute effect sizes (under 0.1) occurred for $AUC2-Ratio$, $Gamma-Ratio$, or $AUC2-Diff$ but many other measures exhibited effect sizes in the small-to-medium range.

Figure 8. Correlation with absolute response bias. Average correlation between estimated metacognitive ability and absolute response bias (i.e., $|c|$) for all 17 measures. As can be seen from the figure, all relationships are relatively small, but there is still a fair amount of uncertainty about each precise value. Error bars show SEM across the 10 subjects in the Locke dataset.

Relevant sections of the Results (all substantially edited)

Dependence on task performance

The results showed that all traditional measures that are not normalized in any way (i.e., *meta-d'*, *AUC2*, *Gamma*, *Phi*, and *ΔConf*) are strongly dependent on task performance: they all substantially increase as the task becomes easier ($p < .001$ for all five measures and three datasets; Figure 3). Critically, the increase across the five measures from the most difficult to the easiest had a very large effect size (Cohen's $d = 2.41, 2.24, 2.64, 1.53,$ and 1.80 for each of the five measures after averaging across the four datasets; Figure 4).

Having established that these five measures strongly depend on task performance, I then examined whether normalizing them removes this dependence. The more popular method of normalization – the ratio method – indeed performed well. The average effect size (Cohen's d) for *M-Ratio*, *AUC2-Ratio*, *Gamma-Ratio*, *Phi-Ratio*, and *ΔConf-Ratio* was $-0.18, -0.39, -0.11, -.17,$ and $-.23$, respectively. These are small effect sizes, except for *AUC2-Ratio* which has medium effect size. Nevertheless, it should be noted that the negative direction of the effect between task performance on metacognitive scores was consistent across all five measures and three datasets (with 9/15 tests being significant at $p < .05$). Thus, while all ratio measures perform much better than the original metrics they are derived from, they tend to slightly overcorrect.

The five difference measures (*M-Diff*, *AUC2-Diff*, *Gamma-Diff*, *Phi-Diff*, and *ΔConf-Diff*) were much less effective in removing the dependence on task performance compared to their ratio counterparts. Indeed, they all exhibited an over-correction where easier conditions led to lower scores with medium average Cohen's d effect sizes (*M-Diff*: -0.58 ; *AUC2-Diff*: -0.49 ; *Gamma-Diff*: -0.39 ; *Phi-Diff*: -0.30 ; *ΔConf-Diff*: -0.55). Further, the relationship between task performance and the metacognitive score was significantly negative for all five measures and three datasets (all 15 p 's $< .05$). These results demonstrate that the difference measures uniformly fail at their main purpose, which is to remove the dependence of metacognitive measures on task performance.

Finally, the two model-based measures (*meta-noise* and *meta-uncertainty*) showed relatively weak but still systematic relationships with task difficulty. Specifically, *meta-noise* decreased for easier conditions in all three datasets (average Cohen's $d = -.29$), whereas *meta-uncertainty* increased for easier conditions in all three datasets (average Cohen's $d = .06$). Both effects were associated with relatively small Cohen's d effect sizes that were comparable to what was observed for the ratio measures. As such, both model-based measures perform as well as the ratio measures in controlling for task performance. Given that *meta-uncertainty* corrected in the opposite direction of the other viable measures (the ratio measures and *meta-noise*) and had the lowest absolute Cohen's d , studies that feature task performance confounds may benefit from performing analyses using both *meta-uncertainty* and at least one more measure.

...

Dependence on metacognitive bias

The results demonstrated that *meta-d'*, *AUC2*, *Phi*, and $\Delta Conf$ tend to increase with higher average confidence, whereas *Gamma* tends to decrease (Figure 5). The average (across the three datasets) Cohen's d effect size was in the medium-to-large range for all five measures (*meta-d'*: 0.44; *AUC2*: 0.51; *Gamma*: -0.61; *Phi*: 0.81; $\Delta Conf$: 0.54; Figure 6). In other words, all five non-normalized measures of metacognition depend on metacognitive bias. All five ratio measures had a positive relationship with metacognitive bias but with smaller Cohen's d effect sizes (*M-Ratio*: 0.27; *AUC2-Ratio*: 0.09; *Gamma-Ratio*: 0.001; *Phi-Ratio*: 0.23; $\Delta Conf$ -Ratio: 0.42). Difference measures performed similarly to ratio measures (*M-Diff*: 0.43; *AUC2-Diff*: 0.10; *Gamma-Diff*: 0.24; *Phi-Diff*: 0.11; $\Delta Conf$ -Diff: 0.34). Finally, the two model-based measures performed similar to the ratio and difference measures and exhibited low-to-medium effect sizes that again went in opposite directions of each other (*meta-noise*: -0.21; *meta-uncertainty*: 0.27). Note that the scores after recoding were similar but slightly larger than the original metacognitive scores before recoding (Figure S3). Overall, researchers who want to control for metacognitive bias would appear to do best if they used *AUC2-Ratio*, *Gamma-Ratio*, *AUC2-Diff*, or *Phi-Diff* as these all featured absolute effect sizes under 0.15. Nevertheless, given that *meta-noise* corrected in the opposite direction of the ratio and difference measures, it may be advisable for results obtained using one of those metrics to be reproduced with *meta-noise*.

...

Dependence on response bias

The results suggested that none of the 17 measures of metacognition are strongly influenced by response bias (Figure 7). Indeed, while a repeated measures ANOVA revealed a very strong effect of condition on response criterion ($F(6,54) = 12.18, p = 1.3 \times 10^{-8}$), it showed no significant effect of condition on any of the measures of metacognition (all p 's $> .13$). Critically, I computed the correlation between the estimated metacognitive ability for each of the 17 measures and the absolute value of the response bias (i.e., $|c|$). The idea behind this analysis is to investigate whether more extreme response bias (either positive or negative) is associated with increases or decreases in estimated metacognitive ability. The results demonstrated that all correlation coefficients were very small (all r 's were between -0.04 and 0.21; Figure 8). There was a fair amount of uncertainty about these values, as seen by the wide error bars in Figure 8, so it is possible some of these relationships may be stronger than the current data suggest. Overall, these results should be interpreted with caution given the small sample size and the fact that a 2-point confidence scale may be noisier for estimating metacognitive scores. Nonetheless, these initial findings suggest that response bias may not have a large biasing effect on measures of metacognition.

Discussion

Alternative ways of quantifying dependence on nuisance variables

I quantified the dependence on nuisance factors by examining effect sizes (Cohen's d and r -values). Alternative ways of examining the dependence on nuisance variables make it difficult to compare between the measures. For example, the difference or ratio of the raw values across easy vs. difficult conditions is not readily comparable across metrics that take different ranges. The main limitation of the

approach I adopted (examining effect sizes) is that noisier measures will have an advantage. In practice, the precision analysis found that 16 of the 17 measures examined here have a similar level of precision, and thus do not substantially differ in their noisiness. Nevertheless, it is possible that the relatively low dependence of *meta-uncertainty* on nuisance variables is in part due to its lower precision (higher noisiness).

2) The main concept of “metacognitive ability” could be clarified. The core idea seems to be that there is a stable trait in individuals that will endure across days at least (as implied by the test-retest reliability analysis), but beyond this I wasn’t sure how the reader should conceive of metacognitive ability. Is it also expected to be stable for an individual across tasks, modalities, cognitive domains, etc.? If so, what is the hypothesized basis of the trait? For example, it might depend on the integrity of a particular neural circuit or be derived from other personality traits such as conscientiousness. Providing evidence on these questions is of course well beyond the scope of this study, but an early statement defining metacognitive ability (at least the author’s working model of it) would help readers who aren’t already immersed in this literature. Somewhat related, I was initially confused by the treatment of task difficulty effects as a “nuisance variable” given its apparently dominating effect on metacognitive performance. In many practical settings, if most of the variance in observed metacognitive performance is explained by a person’s task performance rather than their trait metacognitive ability, then isn’t the observable metacognitive performance (rather some latent ‘ability’) what matters? Eventually, I understood that task performance is a nuisance variable in relation to estimating a hypothesized stable trait ability, but I would’ve understood that perspective earlier with a clearer statement of what this trait might be.

I agree with the reviewer that clarifying the concept of metacognitive ability early on is important to set the stage for the rest of the manuscript. I have now significantly expanded the opening of the Introduction to introduce the concept more fully. As the reviewer suspects, I and others think of metacognitive ability as a reasonably stable trait that is expected to correlate with other stable individual differences, such as brain structure. I also briefly discuss the issue of domain generality, which is indeed usually assumed, though the research on this topic remains hotly debated. I believe that the updated introduction does a much better job at setting the stage for the rest of the paper. I paste the new opening below:

Metacognition is classically defined as knowing about knowing¹. Within this broad construct, the term “metacognitive ability” refers more narrowly to the capacity to evaluate one’s decisions by distinguishing between correct and incorrect answers^{2,3}. High metacognitive ability allows us to have high confidence when we are correct but low confidence when we are wrong. Conversely, low metacognitive ability impairs the capacity of confidence ratings to distinguish between instances when we are correct or wrong. Metacognitive ability is thus a critical capacity in human beings linked to our ability to learn⁴, make good decisions⁵, interact with others⁶, and know ourselves⁷. As such, it is critical that we have the tools to precisely measure metacognitive ability in human participants.

Metacognitive ability is typically assumed to be a somewhat stable trait with meaningful variability across people^{2,8,9}. Consequently, metacognitive ability has been correlated with other stable individual differences, such as brain structure^{10–13}. While metacognitive ability is often assumed to be domain-general and rely on shared neural substrates, this question remains hotly debated^{14–17}. The construct of

metacognitive ability is also thought to be different from other constructs such as task skill or bias, so it is often desirable to find metrics of metacognitive ability unrelated to these other constructs¹⁸.

Below, I first examine the properties that one may desire in a measure of metacognition and then review the known properties of existing measures of metacognitive ability. This brief overview demonstrates that there is little we firmly know about the properties of existing measures of metacognition. The rest of the paper aims to fill this gap by providing a comprehensive test of the critical properties of many common measures of metacognition.

3) The manuscript would benefit from some clarification about the relationship between metacognitive bias and ability. Currently they are treated as separate, with bias characterized as a nuisance variable, but I am not sure they can be fully distinct. For example, if a person is truly extremely biased, for example in terms of being consistently highly confident, then aren't they necessarily also low in metacognitive ability?

I agree with the reviewer that further clarification is needed to elucidate the relationship between metacognitive bias and ability. While this distinction has existed in the literature for a long time, the current terminology was introduced by Fleming and Lau in their seminal 2014 paper (Fleming & Lau, 2014). As they explain, the distinction is meant to mimic the distinction between sensitivity and bias in signal detection theory (SDT).

Specifically, SDT developed measures of sensitivity (d') and bias (criterion c) that are mathematically independent of each other. This allows researchers to estimate sensitivity independent of how biased a person is, which is not true for simpler measures such as percent correct. The conceptual issue raised by the reviewer applies to traditional SDT analyses too: in the presence of extreme bias towards one stimulus category, is it still meaningful to estimate a high d' ? For example, imagine a person with the following pattern of responses: when presented with 100 trials from category A the person picks A 99 times and B 1 time, whereas when presented with 100 trials from category B the person picks A 79 times and B 21 times. This person is extremely biased. Further, their accuracy is only 60%. Nevertheless, the d' for this person is $d' = z(.99) - z(.79) = 2.33 - 0.81 = 1.52$, which is quite high (in the absence of response bias, d' of 1.52 corresponds to about 78% accuracy). Should one still take this d' value seriously?

This is a tough conceptual question that doesn't necessarily have a clear answer. Nevertheless, the standard practice in psychophysics is to simply estimate d' as a measure of sensitivity, with its independence of criterion c being seen as a strength rather than a weakness. The underlying assumption is that one could change their criterion if they wished, which will strongly affect their percent of correct responses but won't change d' . Empirical work on this issue done over the last 50 years mostly supports SDT in that d' does mostly stay stable even for large manipulations of the criterion. So, while doubts in specific cases could remain, the standard practice in psychophysics of trusting the d' value regardless of the strength of the bias seems mostly justified empirically.

The logic with metacognitive bias vs. ability is equivalent to these considerations within SDT. So, while I agree that there's a meaningful conceptual discussion to have, I also believe that it is best for the field to follow the logic enshrined in SDT, where bias and ability are seen as separate concepts that should be independent of each other. I now clarify this theoretical position in the Introduction section, while also making clear that this is simply the perspective taken in the current manuscript and in previous work in the field. I paste the relevant section of the Introduction below:

A second nuisance variable is metacognitive bias, the tendency of people to be biased towards the lower or upper ranges of the confidence scale^{26,27}. This variable can be quantified simply as the average confidence across all trials. Metacognitive bias is under strategic control in that participants can freely choose to use lower or higher confidence. As such, measures of metacognitive ability should ideally remain independent of metacognitive bias because we do not want to measure different ability if people purposefully choose to use predominantly low or high confidence ratings³. The logic here is similar to the logic in SDT, where the measure of performance (d') is designed to be mathematically independent from the measure of response bias (c)²⁸. In the case of SDT, we interpret high d' values as showing high ability to perform the task even if the participant exhibits an extreme bias and, consequently, low percent of correct responses. Similarly, this paper, following the standard in the field¹⁸, adopts the perspective that measures of metacognitive ability should be independent of metacognitive bias.

Beyond this theoretical perspective, there are also practical aspects of this question. The method used to test metacognitive biases, described on p.53, involves recoding confidence values, for example, by recoding all values from 2 to n-1 scale as having the value n-1 (i.e. collapsing values 1 and 2 together). This “bias” should necessarily reduce metacognitive ability to the degree that there is meaningful difference between ratings of 1 and 2, yet the analysis seems to be based on the idea that a good measure will be insensitive to this collapse. Clarification of this would be helpful. Then in Figure 3 it would be useful to see the estimated metacognitive ability values in the original data (prior to recoding). I assume they are all higher. Where “The results demonstrate that meta- d' , AUC2, Phi, and Δ Conf tend to increase with higher average confidence”, is the reason that the tasks were hard, so participants were mostly using low confidence values, so that collapsing values at this end of the scale had a bigger impact?

The reviewer brings another important question. I agree that this is an issue that should be discussed in the manuscript. That said, I believe that it is an open question whether recoding should reduce or increase the estimated metacognitive ability. At least in the case of the two model-based measures (meta-noise and meta-uncertainty), recoding should in theory have no effect on the metacognitive scores (assuming the underlying process models are correct). Let me try to explain why this should be the case for the meta-noise measure derived from the lognormal meta noise model (Shekhar & Rahnev, 2021). According to that model, metacognitive noise can be thought of as variability in the confidence criteria. The amount of variability in each criterion is assumed to be a lognormal function such that the variability increases away from the decision criterion. If the model is *exactly correct* and one has infinite data, then the meta-noise parameter can be equally estimated using 1, 2, 4, or 100 confidence criteria (corresponding to 2-, 3-, 5-, and 101-point confidence scales). It simply doesn't matter

how the experimenter has collected their data. Further, recoding the confidence to move from n -point to $(n-1)$ -point confidence would have no effect on the estimated metacognitive noise as all this does is remove one data point (one criterion) from the data without affecting the rest of the data. But in the case of an exactly correct model and noiseless data (an ideal case not achievable in practice), the experimenter only needs one confidence criterion to estimate meta-noise and having more confidence criteria should not affect the estimated meta-noise either up or down. The same considerations apply to meta-uncertainty derived from the CASANDRE model from Boundy-Singer et al. (2023). This argument is more difficult to make for the remaining 15 measures because they are not derived from process models, but the same logic should still apply.

That said, it is an empirical question what happens in real data. I'm not aware of papers that have examined this question. Consequently, I have followed the reviewer's suggestion to plot the estimated metacognitive ability prior to recoding in the same graph as the estimated values after recoding. In a way, these analyses represent what may be the first empirical investigation into whether coarser vs. finer confidence scales affect the estimated metacognitive ability. I find that there is only a slight effect such that the estimated metacognitive scores after recoding are slightly *higher* than the raw scores before recoding. These results vary by dataset, measure, and subject. To give an overall idea of the effects, I examined whether the average score after the two types of confidence recoding is lower or higher compared to the value before recoding. I did so for each subject and each measure separately. I found that on average 60% (Haddara dataset), 55% (Maniscalco dataset), and 52% (Shekhar dataset) of metacognitive scores were *higher* after recoding compared to the original scores before recoding. Thus, the metacognitive scores after recoding appear to slightly increase on average, but this effect is quite small.

Because this question is not central to the paper, I don't feel that a lengthy discussion of it belongs to the main paper. However, I have now included a new supplementary figure (Figure S3) that shows the plot suggested by the reviewer and reports the above results in the caption. I believe that a more in-depth investigation of how the re-coding affects metacognitive scores compared to the original data before recoding should be conducted in separate research specifically focused on this issue. Below, I paste the new Figure S3:

Figure S3. Comparison of metacognitive scores after recoding using the Xue et al. method with the raw scores before recoding. The figure shows the same data as Figure 5 from the main paper. Critically, it also displays (in thick black lines) the raw metacognitive scores before recoding using the Xue et al. method. As can be seen in the figure, the recoding did not have a strong effect on metacognitive scores relative to the metacognitive scores before recoding. Across all subjects and all 17 measures, I found that on average 60% (Haddara dataset), 55% (Maniscalco dataset), and 52% (Shekhar dataset) of metacognitive scores were higher after recoding compared to the original scores before recoding. Thus, the metacognitive scores after recoding appear to be slightly higher on average, but this effect is quite small. Future research should address what factors may have led to this effect. Error bars show SEM. ***, $p < .001$; **, $p < .01$; *, $p < .05$; ns, not significant.

Minor comments

p.3. Metacognition is more than confidence in decisions. Metacognition has been widely studied in terms of learning, memory, problem solving, everyday abilities, among others. Is the analysis extendable to these?

I agree with the reviewer and now explicitly state that the term metacognition is wide and extends well beyond confidence in decisions. The current analyses are certainly extendable to additional domains, but since I only examine perception datasets, it remains an open question whether the results may change when other domains are examined. In addition, most measures examined here only apply for paradigms that include a binary decision with confidence because most of the measures are built on top of SDT, which only operates in the context of 2-choice tasks. I now discuss both issues under a “Limitations” sub-heading in the Discussion and paste the relevant text below:

Limitations

The present work has several limitations. ... Fourth, the current analyses were conducted exclusively in the context of perception. Metacognition has been widely studied in the context of learning, memory, problem solving, etc¹. While the results here are expected to generalize to these other domains, additional research is needed to confirm that. Fifth, most measures examined here only apply to 2-choice tasks and thus cannot be used for designs with estimation tasks, n-choice tasks, etc.

p.4. Is really true that validity has been ignored? Most/all measures in the literature assess the degree to which metacognitive evaluations match objective reality. Isn't that a face valid approach?

I agree that the existing measures of metacognition have inherent face validity, which is also likely why a more systematic investigation into their validity has not been undertaken before. I now clarify this point in the text and paste the updated text below:

The most important property of any measure is that it is valid: namely, it should measure what it purports to measure¹⁹. Existing measures of metacognitive ability assess the degree to which confidence is associated with objective reality, thus making them face valid. Still, we lack a formal way of verifying the validity of existing measures.

p.11. Do the analyses listed in Table 4 correspond to the ones originally conducted by the study authors, or are they the ones done here?

The analyses listed in Table 4 are the ones done in the current paper. In no case did the original study perform these same analyses. I have now clarified this in both the text and the table caption, and paste the resulting text below:

Introduction

In addition, Table 4 indicates the dataset(s) used for each type of analysis in the current paper.

Table caption

Table 4. Datasets used in the current paper. The table lists details of each dataset and indicates which analyses in the present paper each dataset was used for.

p.17. The pairwise comparisons should be post hoc tests, because they are run based on inspection of the data. Also here, can the author provide any intuition for why meta-uncertainty does so much worse?

I agree with the reviewer and now report both uncorrected statistical tests and tests corrected for the presence of $17 \cdot 16 / 2 = 136$ pairwise comparisons. I also agree with the reviewer that it would be helpful to provide an intuition about why meta-uncertainty does so much worse than the other measures. The reason why I refrained from doing so in the previous version of the

manuscript is that the reasoning would necessarily be speculative as it is difficult to be sure. Nevertheless, I suspect that the issue comes from the fact that meta-uncertainty is based on a complex model-fitting process, which may introduce additional noise in the estimation process. I paste below the updated paragraph from the Results section:

The one exception was *meta-uncertainty*, which had substantially lower average precision score in both the Haddara (*meta-uncertainty*: 0.37; average of other measures: 0.67) and the Maniscalco datasets (*meta-uncertainty*: 0.30; average of other measures: 0.53). Moreover, pairwise comparisons showed that the precision for *meta-uncertainty* was lower than every one of the other 16 measures in both datasets before multiple comparison correction (p 's < .05 for all 32 comparisons). In the Haddara dataset, 15 of the 16 comparisons remained significant even after applying a very conservative Bonferroni correction for the existence of $\frac{17 \times 16}{2} = 136$ pairwise comparisons; in the much smaller Maniscalco dataset, no comparison remained significant after this conservative correction. This difference between *meta-uncertainty* and the remaining measures may stem from the noisiness of the process of estimating *meta-uncertainty* in the presence of relatively few trials. In fact, the original authors who introduced *meta-uncertainty* already warned about the dangers of trying to compute this variable using low trial numbers³⁵.

p.20. “The reason why AUC2-Ratio does not perform well is that AUC2, unlike the other four measures here, has a lower boundary of 0.5 rather than 0.” Please explain why this feature leads to poorer performance, because it is not intuitively clear.

After trying to formalize my intuition about this, I realized that my initial intuition about the lowest boundary being the main issue for the worse AUC-Ratio performance (compared to other ratio measures) may have been incorrect or at least incomplete. It’s also not clear to me anymore that there is a fundamental difference between AUC-Ratio and the other Ratio measures as the new Figure 4 shows. Consequently, I’ve decided to remove the statement above and instead emphasize the commonality across the ratio measures. I paste again the new Figure 4 below, which suggests that all ratio measures are rather similar to each other:

p.23. “Given that meta-noise corrected in the opposite direction of the other five measures, it may be advisable for any result to be reproduced both meta-noise and at least one more measure from the list above”. It would greatly strengthen this recommendation if it was explained why meta-noise behaved in this way, so that the reader is reassured that the opposite direction is a feature not a bug of this method.

I agree with the reviewer and in some cases, it is possible to pinpoint why a certain measure shows a particular effect. In this specific case, however, I’m not sure why meta-noise behaves in this way. The measures meta-noise and meta-uncertainty are derived from process models that postulate specific computations for confidence. In those computations perfectly match reality, then the corresponding measure should be independent of any nuisance variables. The fact that they show some dependencies means that their assumed process models deviate from reality in some way(s). However, currently we don’t have a model of confidence that consistently fits better than the models from which meta-noise and meta-uncertainty are derived, so we don’t know the ground truth. Once such a model is constructed, it would hopefully allow us to understand why exactly meta-noise and meta-uncertainty behave the way they do. In the meantime, all I can do to address this comment is to acknowledge the fact that we don’t know why the effects I’m observing are the way they are. Below, I’m pasting a new part of the Discussion that examines these issues:

Several measures showed dependence on nuisance variables that went in the opposite direction from most other measures (*meta-uncertainty* for task performance, as well as *meta-noise* and *Gamma* for metacognitive bias). As such, these measures may be especially useful to use when there is a concern that results may be driven by a specific nuisance variable. Unfortunately, it is currently difficult to determine why these measures show the opposite effects (or, for that matter, why most measures show the dependencies they show). Understanding the nature of these relationships will likely require further progress in developing well-fitting process models of metacognition^{53,54}.

p.46. For readers unfamiliar with the original work, it would be helpful to give an intuition for why a lognormal distribution is used in the meta-noise model, and what it implies about metacognition. Similarly for the meta-uncertainty model, it would be useful to give an indication of how uncertainty about uncertainty affects metacognitive reports (and/or how this is estimated).

I agree and have added a few sentences that discuss in slightly more detail the models from which meta-noise and meta-uncertainty are derived:

Shekhar and Rahnev proposed the lognormal meta noise model, which is an SDT model with the additional assumption of lognormally distributed metacognitive noise that affects the confidence criteria. The lognormal distribution was used because it avoids non-sensical situations where a confidence criterion moves on the other side of the decision criterion. The metacognitive noise parameter (σ_{meta} , referred here as *meta-noise*) can be used as a measure of metacognitive ability.

...

The measure developed by Boundy-Singer³⁵ - *meta-uncertainty* – is based on a different process model of metacognition, CASANDRE, that implements the notion that people are uncertain about the uncertainty in their internal representations. Specifically, it denotes the noise present in the estimation of the sensory noise. The second-order uncertainty parameter, *meta-uncertainty*, represents another possible measure of metacognition.

p.51. “If multiple difficulty conditions were present in a given dataset, I performed a linear regression on the values of each measure and then ran a one-sample t-test against 0 on the resulting slopes.” Please clarify the description of the analysis. Is a separate regression fit per participant and then second-order statistics performed on the resulting regression betas?

Yes, I performed a separate regression fit per participant, followed by second-order statistics on the regression betas. However, as this analysis pertained to the Adler dataset (which has now been removed), I have now simply deleted this text.

p.53. “In the second recoding, only the rating of n is decreased by one, resulting in a bias towards high confidence.” Wouldn’t this manipulation decrease confidence? Is the idea that this is equivalent to collapsing at the high end of the scale?

I thank the reviewer for pointing out the lack of clarity in the way I phrased this. I meant that the second recoding results in a bias towards high confidence *relative to the first recoding* (not relative to no recoding). I have now re-written the few sentences in that portion of the manuscript to clarify this. I paste them below (together with Figure 5 for easy reference):

Specifically, an n-point scale is transformed into an (n-1)-point scale in two ways. In the first recoding, the ratings from 2 to n are all decreased by one. In the second recoding, only the rating of n is decreased by one. When compared to each other, the first method results in a bias towards lower confidence compared to the second method (see mean confidence values in the bottom right of Figure 5).

Typos and language

Check and correct formatting of in-text citations such as “To address this issue, (Maniscalco & Lau, 2012) developed...””.

I have corrected all citations and also converted the citation style into the one required by Nature Communications.

p.11. “Dependance” in Table 4 but “dependence” elsewhere.

Thanks for catching that, I’ve now used “dependence” everywhere, including Table 4.

p.16, Figure 1. The panels aren’t labeled in a way that matches the figure caption. What are the horizontal lines in the plots of “Decrease in SD units”?

Thank you for pointing out this error. I have now added panel letters in the figure, so that it matches the figure caption. The horizontal lines represent the average value of the first 16 measures and are added to make it easier to see any differences between those measures. I have now added this information in the figure caption. I’m pasting the updated figure and figure caption below:

Figure 1. Validity and precision of each measure. Results of an artificial corruption of the confidence ratings where confidence for correct trials was decreased by 1, and confidence for incorrect trials was increased by 1. (A) Detailed results for the Haddara dataset. Each one of the 17 measures of metacognition showed a decrease with this manipulation. The plot shows the decrease in units of the standard deviation (SD) of the measure’s fluctuations across different bins. The decrease was computed for bin sizes of 50, 100, 200, and 400 trials, as well as for 2, 4, and 6% of trials being corrupted. (B,C) Average precision in SD units for each measure in the Haddara and Maniscalco datasets averaged across the four bin sizes and the three levels of corruption. The horizontal lines represent the average value of the first 16 measures and are added to make it easier to see any differences between those measures. Error bars show SEM.

p.31. Figure 6 shouldn’t be a line graph because neighboring points aren’t theoretically linked any more than non-neighboring ones.

I agree and have now removed the connecting lines in the figure. I've also added ICC values in addition to the originally reported Pearson correlations. I'm pasting the updated figure below:

Figure 10. Test-retest reliability of metacognitive scores. Test-retest correlations in the Haddara dataset (6 days, 500 trials per day, 70 subjects) show generally low test-retest reliability. Upper panel shows ICC values, whereas lower panel shows Pearson correlation. The test-retest reliability was low-to-moderate for the measures *meta-d'*, *AUC2*, *Gamma*, *Phi*, and *ΔConf* and very low for the remaining measures.

p.46. “(Kornell et al., 2007)”

Fixed.

Reviewer 2

Review of manuscript NCOMMS-23-32257 titled “Measuring metacognition: A comprehensive assessment of current methods”.

In this manuscript the author compares several methods for measuring metacognitive sensitivity across several public data sets. The manuscript is well written and will be useful for researchers studying metacognition. I present below my recommendations for improving the manuscript in what is my standard review structure.

I thank the reviewer for the constructive comments and address all recommendations below.

1. Do the conclusions follow from the data?

The author provides many useful tests of current measures of metacognition, with useful recommendations for moving forward.

Thank you.

2. Is the analysis conducted appropriately?

When testing for reliability in test-retest or split-half data, an absolute agreement intraclass correlation coefficient (ICC; Shrout & Fleiss, 1979) is more appropriate than the Pearson correlation coefficient. The Pearson correlation coefficient tests $y = mx + b$, how well x predict y , allowing x and y to have different means (differing by b). The absolute agreement ICC, as the name suggests, tests $y = x$, how well x predicts y assuming they should be the same. The author may wish to keep the Pearson's r correlation coefficient for comparison with Guggenmos 2021 (or use the ICC and the normalised mean absolute error, the other reliability measure used in Guggenmos, 2021). See Matheson (2019) for more detailed explanation of ICC in test-retest contexts.

I thank the reviewer for pointing out the fact that ICC is more appropriate here. I have now computed ICC scores. As recommended by the reviewer, I've also kept the Pearson's correlation values for ease of comparison with Guggenmos (2021). The results of the two metrics are very similar and there are slight differences only for a few measures in ways that don't alter the conclusions. I'm pasting the new figure below:

Figure 10. Test-retest reliability of metacognitive scores. Test-retest correlations in the Haddara dataset (6 days, 500 trials per day, 70 subjects) show generally low test-retest reliability. Upper panel shows ICC values, whereas lower panel shows Pearson correlation. The test-retest reliability was low-to-moderate for the measures *meta-d'*, *AUC2*, *Gamma*, *Phi*, and *ΔConf* and very low for the remaining measures.

A Fisher transform should be applied to correlation coefficients prior to averaging. Correlation coefficients are bounded and therefore not normally distributed. Toward the bounds the distributions become skewed and therefore the mean is inappropriate. The Fisher transform gives a normally distributed scalar of correlation coefficients (the inverse can be applied to the average).

I agree with the reviewer. In fact, I applied a Fisher transform for all analyses. Generally, I computed a set of r-values, then Fisher-transformed them, averaged the resulting z-scores, and finally converted the z-score back into an r-score for plotting. However, I realize that this procedure was not sufficiently clarified. I've now clarified this process in the Methods and paste the resulting text below:

Assessing split-half reliability

... The obtained r-values were then z-transformed, averaged, and the resulting average z value was transformed back to an r-value for reporting and plotting purposes.

Assessing test-retest reliability

... As with the split-half analyses, the obtained correlation coefficients were then z-transformed, averaged, and the resulting average z value was transformed back to a correlation coefficient (ICC or r-value) for reporting and plotting purposes.

The analysis referring to ‘task-performance’ actually only indirectly applies to performance via externally manipulated difficulty. The author should rephrase to refer to task-difficulty throughout, or perform the analysis on task-performance (d'). An analysis of task difficulty would make the comparisons across data-sets better, and be more direct (for example, difficulty levels 5 and 6 of the Adler data don't seem to differ in terms of task performance). The author may also note that some readers may be confused by the x-axis of Figure 2 in that increasing numerical label of difficulty corresponds to increasing performance, presumably because it is actually decreased difficulty.

I agree with the reviewer and have consequently changed the figure (now Figure 3) to plot task performance (d') on the x axis. I believe that this is more informative and more intuitive, as noted by the reviewer. In addition, based on a comment by Reviewer 1, I have decided to remove the Adler dataset due to its small sample size and complex design, which may have introduced spurious effects. I paste the new Figure 3 and its caption below:

Figure 3. Dependence on task performance. Estimated metacognitive ability for all 17 measures, as well as d' , criterion, and confidence for different difficulty levels in four datasets (Adler, Shekhar, Rouault1, and Rouault2). Traditional measures of metacognition (top row) all showed a strong positive relationship

with task performance, whereas all Diff measures (third row) show a strong negative relationship. Ratio measures (second row) and the two model-based measures (meta-noise and meta-uncertainty) performed much better but still showed weak relationships with task performance. Note that higher numbers on the x axis indicate easier conditions. ***, $p < .001$; **, $p < .01$; *, $p < .05$; ns, not significant.

I found myself wondering if the ratio measures are comparable? They should be in similar enough units for a Pearson correlation. I think other readers would be interested if the author is willing to add this comparison.

If I understand correctly, the reviewer is suggesting that I correlate the different measures across subjects. I think that this is an excellent suggestion and that it would, in fact, be useful to extend this analysis to all 17 measures. I've now performed these correlations and plotted them in Figure 11. As the figure shows, the ratio measures are indeed highly correlated. In addition, there is very strong correlation between all ratio and difference measures, suggesting that the SDT-based normalization makes them behave similarly. Below, I paste the new section of the Results that presents and discusses these results in detail:

Across-subject correlations between different measures

Lastly, I examined how different measures are related to each other by performing across-subject correlations. Note that these analyses should be interpreted with extreme caution because the correlation between two measures could be driven by a third factor. For these analyses, I again used the Haddara (3,000 trials per subject), Maniscalco (1,000 trials per subject), and Shekhar (3 difficulty levels, 2,800 trials per subject) datasets. As in previous analyses, I examined each difficulty level in the Shekhar dataset in isolation and then averaged the results across the three difficulty levels. For each dataset, I computed each measure of metacognition based on all trials in the experiment and examined the across-subject correlations between different measures.

Overall, the 17 measures of metacognition showed medium sized across-subject correlations with each other (average $r = .49, .55,$ and $.56$ for the Haddara, Maniscalco, and Shekhar datasets, respectively; Figure 11). These analyses seemed to reveal three groups of measures. The first group consists of the five non-normalized measures (*meta-d'*, *AUC2*, *Gamma*, *Phi*, and $\Delta Conf$), which exhibited average inter-measures correlation of $.60$ ($r = 0.60, 0.63,$ and 0.58 in each dataset). The second group consists of the five ratio and five difference measures, which exhibited average inter-measures correlation of $.63$ ($r = 0.62, 0.62,$ and 0.63 in each dataset). The average correlation between the first two groups of measures was slightly weaker than the within-group correlations ($r = .51$ on average; $r = 0.42, 0.55,$ and 0.55 in each dataset). Note that these results could be driven by the fact that all five non-normalized measures are strongly driven by d' , thus increasing the correlations between them. It may also be that the SDT-based normalization makes all ratio and difference measures similar to each other.

Figure 11. Across-subject correlations between different measures. The figure depicts *r*-values obtained from conducting across-subject Pearson correlations between all pairs of measures. The analyses were conducted separately for the Haddara, Maniscalco, and Shekhar datasets. Star symbols indicate significant correlation at the $p < .05$ uncorrected level.

Finally, the third group of measures consists of the two model-based measures, which showed the strongest divergence from the rest of the measures. Specifically, *meta-noise* had an average correlation of 0.35 with the remaining measures ($r = 0.35, 0.34,$ and 0.37 in each dataset) and *meta-uncertainty* had an average correlation of 0.44 with the remaining measures ($r = 0.33, 0.45,$ and 0.53 in each dataset). The measures *meta-noise* and *meta-uncertainty* had a very weak correlation between each other ($r = 0.15, 0.03,$ and 0.06 in each dataset). These results suggest that the two model-based measures may capture unique variance related to metacognitive ability.

Contrary to the reporting summary, the test statistics, confidence intervals and degrees of freedom are not disclosed for the tests of dependence on ‘task performance’, dependence on metacognitive bias, and dependence on response bias (could give the range of values in cases where many tests were performed).

I struggled with this comment a lot. I understand the need for detail and transparency and fully support that in my work. However, when I experimented with reporting ranges of values, this created text that was extremely complex and confusing. I also checked previous papers and it seems that the practice of reporting only *p*-values when multiple tests are mentioned is standard for Nature Communications. Conversely, I couldn’t find examples of papers that report ranges of *t*-values, *p*-values, and CIs in a similar context as the statistics in the current paper (though I’m sure that example of this must exist). I have therefore kept the previous way of reporting some of the statistics, though I now de-emphasize significance testing and base all conclusions on effect sizes instead. I have added a statement about it in a new section in the Methods. If the reviewer and Editor feel that this approach is unacceptable, I’m prepared to remove the statistical testing from the paper. Alternatively, I would also appreciate any examples of how to report such statistics without compromising the readability of the paper. I’m pasting below the new section in the Methods where I address the issue of only reporting *p*-values when multiple tests are reported:

All conclusions in the paper are based on effect sizes (Cohen’s *d*, *r*, and ICC values). However, for completeness, I sometimes refer to the results of null-hypothesis statistical tests. As is standard practice, in cases where I report the results of multiple tests together, I only include the *p*-values. All remaining

information, such as test statistics and degrees of freedom, can be obtained from the provided analysis codes.

3. Do the introduction and methods give sufficient information for understanding the experiment?

The introduction is well structured and clear (with the exception of the minor issues below). The methods also gives sufficient detail.

Thank you.

4. Does the discussion sufficiently outline the findings?

The discussion is well rounded and also very clear.

Thank you.

Minor:

The definition of metacognition (first sentence of introduction) does not match the reference (who define metacognition as knowing about knowing – the book is largely about metamemory).

I thank the reviewer for pointing out the mismatch here. I have now edited the opening of the Introduction to clarify how Metcalfe and Shimamura define metacognition, and what is meant specifically by “metacognitive ability”. I paste the updated text below:

Metacognition is classically defined as knowing about knowing¹. Within this broad construct, the term “metacognitive ability” refers more narrowly to the capacity to evaluate one’s decisions by distinguishing between correct and incorrect answers^{2,3}.

“Several papers have used simulations to investigate some of the properties of measures of metacognition (Barrett et al., 2013; Guggenmos, 2021), but this approach is potentially problematic because it is a priori unknown how well the process models used to simulate data reflect empirical reality.” This sentence implies that the Guggenmos paper did not examine data also. The author may wish to consider painting it in a more positive light given many of the comparisons made there are duplicated in the current manuscript.

Thank you for pointing this out. To be clear, I think extremely highly of both the Guggenmos and Barrett papers. I also didn’t intend for this sentence to imply that Guggenmos only used simulations, though upon reflection, I see that the sentence comes off this way. To avoid confusion, I have now made this sentence about papers that focus exclusively on simulations and have correspondingly removed the Guggenmos paper from this sentence. I have now edited the sentence to as not to appear to disparage these excellent previous contributions:

Several papers have relied exclusively on simulations to investigate some of the properties of measures of metacognition^{35,36}. Such investigations are important but cannot substitute empirical studies because it is a priori unknown how well the process models used to simulate data reflect empirical reality.

“One paper (Azzopardi & Evans, 2007) examined the properties of a measure, Type-2 d' , which was subsequently shown to be based on faulty assumptions (Galvin et al., 2003)...” 2003 is not subsequent to 2007. Azzopardi and Evans actually investigated Kunimoto’s a' , showing it to be corrupted by response bias. This issue of bias was popular at the time; Persuade and colleagues (2007) published a wagering method (similar to the Kunimoto 2001 method), which was rapidly debunked by Clifford et al. (2008). The Galvin paper is about type-2 ROCs, listing many caveats about confidence ratings that make ROC-like analyses (including meta- d') based on potentially false assumptions. Azzopardi and Evans discuss this in their paper. This is to say, there are interesting things about the references in the sentence, but the sentence is incorrect.

Thank you again, I completely agree. Indeed, Evans and Azzopardi didn’t propose a new measure, but instead examined the properties of Kunimoto’s a' . I’ve now edited this sentence to correctly convey the matter:

Evans and Azzopardi³⁷ empirically showed that a specific measure of metacognition, Kunimoto’s a' ³⁸, exhibits a strong dependence on response bias. Because Kunimoto’s a' is built on wrong distributional assumptions³⁹, it is not investigated here.

“The one exception was meta-uncertainty, which had substantially lower average precision score in both the Haddara (meta-uncertainty: 0.37; average of other measures: 0.67) and the Maniscalco datasets (meta-uncertainty: 0.30; average of other measures: 0.53).” Boundy-Singer et al., 2022 themselves note issues of parameter recovery for small numbers of trials (100) and recommend at least 700 trials for this model. The author may wish to note this (none of these tests were performed according to the recommendations of the authors for the meta-uncertainty model).

I agree with the reviewer that it is important to point out that Boundy-Singer et al. discuss this issue in their paper. I’ve now done so and paste the updated paragraph below:

The one exception was *meta-uncertainty*, which had substantially lower average precision score in both the Haddara (*meta-uncertainty*: 0.37; average of other measures: 0.67; ratio = 0.56) and the Maniscalco datasets (*meta-uncertainty*: 0.30; average of other measures: 0.53; ratio = 0.58). Moreover, pairwise comparisons showed that the precision for *meta-uncertainty* was lower than every one of the other 16 measures in both datasets before multiple comparison correction (p 's < .05 for all 32 comparisons). In the Haddara dataset, 15 of the 16 comparisons remained significant even after applying a very conservative Bonferroni correction for the existence of $\frac{17 \times 16}{2} = 136$ pairwise comparisons; in the much smaller Maniscalco dataset, no comparison remained significant after this conservative correction. This difference between *meta-uncertainty* and the remaining measures may stem from the noisiness of the process of

estimating *meta-uncertainty* in the presence of relatively few trials. In fact, the original authors who introduced *meta-uncertainty* already warned about the dangers of trying to compute this variable using low trial numbers³⁵.

“within-subject studies of metacognition do not require high reliability – a measure that inherently depends on a large spread of scores across subjects” How do within-subjects studies require spread between subjects?

Apologies for the confusion. I can see how this sentence is ambiguous. I did not mean to imply here that within-subject studies require spread between subjects. I meant to say that such spread is irrelevant to within-subject studies but is needed for between-subject studies that rely on high test-retest reliability. I’ve now edited the sentence to clarify this point:

Note that the reliability of a measure is enhanced by both high precision and large spread of scores across subjects, so both of these two factors are important for between-subject analyses. In contrast, within-subject analyses only require high precision. Therefore, low reliability scores are not necessarily problematic for within-subject designs.

“I computed a total of 17 measures of metacognition and provide Matlab code for their estimation” It would be nice to have the link with this sentence (in addition to the end of the document).

I agree and have included a reference to the codes in that sentence.

Reviewer 3

This submission addresses an important topic in contemporary cognitive science – metacognition, and how it should be measured. The authors assess most of the currently available metrics of metacognitive sensitivity, using a sensible set of measures. Overall, they find that most of these existing measures are reasonably interchangeable on these metrics. Whether this leaves the reader feeling comforted or despondent may depend on whether they take a glass half full or brimming perspective of the world, and of this area of research.

Overall, I think this submission is a welcome addition, building considerably on the scope of other recent, similar publications. It will be a welcome addition to the public record.

My only reasonably substantive points relate to terminology, and to requests that the authors consider adding to their discussion to address conceptual issues.

I thank the reviewer for the positive evaluation and for raising these important issues. I address each of these comments when it is brought in more detail below.

Specific Points

1) High precision, or High sensitivity?

The authors refer to High precision as a desirable quality that they are measuring, but to my mind their description and use of this term describes the sensitivity of metacognitive sensitivity tests, not their precision.

To measure 'precision' / sensitivity, the authors add a counter measure to ratings of confidence in data, and quantify how far this diminishes evidence of metacognitive sensitivity relative to unperturbed datasets. This speaks to the sensitivity of the measures (i.e. if they are sensitive to metacognitive sensitivity at all), but not to the precision of these measures (i.e. are they just measuring metacognitive sensitivity, or are they also measuring other related constructs).

I agree with the reviewer about the need for accurate terminology here. I use the term "precision" following its definitions in the literature as "the ability to repeatedly measure a variable with a constant true score and obtain similar results" (Nebe et al., 2023), "the margin of error" in a measurement (Cumming, 2014) or "the spread of values that would be expected across multiple measurement attempts" (Luck et al., 2021). Note that precision, according to this type of definition, does not refer to whether a measure is only affected by the construct of interest. That said, to be able to compare between different measures of metacognitive ability, it is necessary that a measure of precision is standardized across the different measures of metacognition, which is why I quantify precision as the consistency of measurement relative to a manipulation that decreases metacognition. I agree with the reviewer that this last part makes what I'm actually measuring also describable by the term "sensitivity". However, I find that it becomes confusing to talk about the "sensitivity of a measure of metacognitive sensitivity" and therefore prefer to use the term "precision" instead. That being said, I recognize that this is a terminological issue with no ultimate influence on the science. So, if the reviewer still finds that the use of the term "precision" is confusing, I would be happy to make a switch in a further revision. Below I paste the edited part of the Introduction where I now introduce the term precision in more detail and clarify its intended definition for the purposes of the current paper:

A related property is precision. I use the term "precision" following its definitions in the literature as "the ability to repeatedly measure a variable with a constant true score and obtain similar results"²⁰, "the margin of error" in a measurement²¹ or "the spread of values that would be expected across multiple measurement attempts"²². Note that precision here does not refer to whether a measure is only affected by the construct of interest. Precision has been largely ignored in the context of measures of metacognition and we currently lack methods to measure it.

2) Why should measures of metacognitive sensitivity be independent of task difficulty?

The authors assert that a good measure of metacognitive sensitivity should be independent of task difficulty. I question this assertion and cannot see any underlying logic to it.

Cognitive load is a standard manipulation, used to undermine performance in cognitive tasks. Moreover, there does not need to be a great overlap between cognitive operations for a cognitive load (from concurrently performing a difficult cognitive operation) to undermine performance on another task. Similarly, I can see no reason why the demand to perform a difficult cognitive operation could not undermine a person’s ability to attend to the nuances of their feelings of decisional confidence – even if insight into these is somewhat independent of decisions themselves.

Another way to express this concern is that our feelings of confidence could obey a Weber fraction, in which case it would be harder (in absolute terms) to distinguish between feelings of confidence that signal large magnitudes of uncertainty, relative to when we distinguish between feelings of confidence that signal small amounts of uncertainty.

I would encourage the authors to consider these points, and to address them in a revised manuscript.

This is an excellent point also brought up by R1. The reviewer is correct that if difficulty is manipulated by introducing cognitive load or other task demands that may tax the metacognitive system, then one would no longer expect metacognitive ability to remain the same anymore. In fact, there is some empirical work on this question in the context of whether metacognitive ability is affected by working memory load with somewhat mixed results (Konishi et al., 2020, 2021; Maniscalco & Lau, 2015). However, I believe that when difficulty is affected via stimulus rather than task manipulations, then it is more reasonable to expect that metacognitive ability shouldn’t change (see, for example, Fleming & Lau, 2014; Rahnev, 2021). Consider the case of visual perception tasks with confidence. For such tasks, there seems to be little reason to believe that the underlying metacognitive ability should be affected by stimulus contrast. If so, one may want to measure the same metacognitive ability regardless of stimulus contrast in such tasks. Thus, the assumption that task performance shouldn’t affect metacognitive ability seems to apply more readily to stimulus than task manipulations.

That said, even if one does not agree that metacognitive ability should be independent from task performance, examining how each measure depends on task performance is still informative, especially if – as the current paper shows – there are large differences between measures. I have now discussed the issues above in a lot more detail in the Introduction and have also made edits throughout to qualify when the logic of task performance independence is likely to apply. I paste the edited text in the Introduction below:

The nuisance variable that has received the most attention is task performance. It is often desirable that a measure of metacognition should not be affected by whether people happened to be performing an easy or a difficulty task^{3,18}. For example, in visual perception tasks with confidence, there is little reason to believe that the underlying metacognitive ability should be affected by stimulus contrast. Thus, one may want to measure the same metacognitive ability regardless of contrast level. Note that there are subtleties here. If difficulty is manipulated by introducing cognitive load or other task demands that may tax the metacognitive system, then one would not necessarily expect metacognitive ability to remain the

same anymore (though whether metacognitive ability is affected by working memory load remains a topic of debate²³⁻²⁵). Therefore, the logic here applies more readily to stimulus than task manipulations. That said, even if one does not agree that metacognitive ability should be independent from task performance, examining how each measure depends on task performance is still informative, especially if there are meaningful differences between measures. Task performance can be computed as d' , which is a measure of sensitivity derived from signal detection theory (SDT).

3) Of course, all the measures have similar outcomes. They are all very similar.

All the measures the authors consider are variants on signal-detection-theory (SDT) based instruments. The conceptual differences between the measures are very slight, and so it seems reasonable that the different outcomes of the analyses here are similarly slight.

There are increasingly sophisticated process models, some of which the authors themselves have developed, that are more nuanced. Such approaches have found benefit to the presumption of different shaped noise distributions, and conceptually they could offer some insight into the temporal evolution of a decision process. Here the authors want to unfold a standard set of tests, so that each measure can be equally assessed. But this conceptually levelling of the playing field by the development of some generic tests could create a false impression of equivalence. By reducing all accounts to a generic level that can equally be assessed, the authors may have assured that no measure can substantially outperform its SDT basis.

I think the authors should discuss this issue in a revised manuscript.

This is very helpful feedback. In retrospect, I can see that the manuscript gives the impression that all measures have basically similar outcomes. Indeed, the 10 SDT-based measures (5 ratio and 5 difference measures) do behave similarly and that's to be expected. In a way, this is to show that there's nothing particularly special about M-Ratio specifically – other measures constructed using the same logic behave similarly.

That said, there are quite a few meaningful differences between the 5 non-normalized measures (*meta-d'*, *AUC2*, *Gamma*, *Phi*, and $\Delta Conf$), as could be expected given that only *meta-d'* is more directly derived from SDT. The two model-based measures also have many differences from the other measures. However, I see that given that 10 of the measures are quite similar and there are so many measures and analyses, the differences between the remaining measures get lost.

I have addressed this issue in several different ways. First, I emphasize early on (in the end of the Abstract and Introduction) that the results for different measures are not equivalent. Second, I realize that many of the figures are very complex and it's difficult to immediately see if there are meaningful differences in the measures (especially in the case of the nuisance variable analyses). Therefore, I have created new "summary" figures that plot the effect sizes for each nuisance variable analysis. Third, I expand on the most important differences between the different measures in the Discussion.

That being said, I also agree with the reviewer that the requirement to run the same tests on all measures constitutes a conceptual “levelling of the playing field”, which can disadvantage more complex and flexible measures. This is, as noted by the reviewer, especially true for model-based measures that could be applied to a wider set of designs than traditional measures. I have now included a discussion of this point in the Discussion section of the manuscript.

I paste below the relevant parts of the Abstract, the Introduction, the new figures, and the relevant parts of the Discussion:

Abstract

This comprehensive assessment paints a complex picture: no measure of metacognition is perfect and depending on the details of the experiment, different measures may be preferable.

Introduction (last paragraph)

Overall, I find that no current measure of metacognitive ability is “perfect” in the sense of possessing all desirable properties. Nevertheless, they are not equivalent either with many important differences between measures emerging. Based on these results, I make recommendations for the use of different measures of metacognition based on the specific analysis goals.

Results (new figures)

Figure 2. Normalized precision of each measure. Normalized precision for all 17 measures in each of the two datasets. The precision values are normalized such that the average precision level of the first 16 measures equals 1 in each of the two datasets. As can be seen, *meta-uncertainty* has substantially lower level of precision than the rest of the measures. The differences between the remaining measures are not always trivial but tend to be smaller.

Figure 4. Effect sizes for dependence on task performance. Effect sizes (Cohen’s d) is plotted for each metric and dataset. As can be seen in the figure, non-normalized traditional measures (i.e., *meta-d*, *AUC2*, *Gamma*, *Phi*, and *ΔConf*) show strong positive relationship with task performance. Corrections with the Ratio and Diff methods reverse this relationship, with the Ratio correction being clearly superior. The model-based metrics *meta-noise* and *meta-uncertainty* perform well too, with *meta-uncertainty* showing particularly low effect sizes.

Figure 6. Effect sizes for dependence on metacognitive bias. Effect sizes (Cohen’s d) is plotted for each metric and dataset. As can be seen in the figure, all metrics except for *Gamma* and *meta-noise* have a mostly positive relationship with metacognitive bias (i.e., higher confidence leads to higher estimates of metacognition). The smallest absolute effect sizes (under 0.1) occurred for *AUC2-Ratio*, *Gamma-Ratio*, or *AUC2-Diff* but many other measures exhibited effect sizes in the small-to-medium range.

Discussion

Unique advantages and disadvantages of different measures

Several measures feature unique advantages and disadvantages (Figure 12). For example, four of the ratio measures (*M-Ratio*, *Gamma-Ratio*, *Phi-Ratio*, and *ΔConf-Ratio*) become unstable for difficult conditions because they include division by variables (d' , expected Gamma, expected Phi, and expected ΔConf , respectively) that are very close to 0 in such conditions. These measures should therefore be used preferentially when performance levels are relatively high (e.g., one should aim for d' values above 1, which roughly corresponds to accuracy values above 69%).

An advantage of *AUC2*, *Gamma*, *Phi*, and *ΔConf* is that they all work well with continuous confidence scales. All other measures rely on SDT-based computations that necessitate that continuous scales are binned before analyses. Such binning may lead to loss of information, but it is currently unclear how much signal may be lost by different binning methods.

The two model-based measures – *meta-noise* and *meta-uncertainty* – have unique advantages and disadvantages. Their main advantage is that all their underlying assumptions are explicitly known. Conversely, other measures must necessarily include hidden assumptions that are difficult to reveal without linking them to a process model of metacognition³. Another unique advantage of these measures is that they can in principle be applied much more flexibly. For example, when an experiment contains several conditions, other measures do not allow the estimation of a single measure of metacognition and simply ignoring the different conditions can lead to inflated scores⁴⁹. Conversely, both *meta-noise* and *meta-uncertainty* allow different conditions to be modeled as part of their underlying process models and thus a single metacognitive score can be computed in a principled way across many conditions. A possible disadvantage of both measures is that they can only take positive values and therefore cannot be used for situations where metacognition may contain more information than the decision itself.

Several measures showed dependence on nuisance variables that went in the opposite direction from most other measures (*meta-uncertainty* for task performance, as well as *meta-noise* and *Gamma* for metacognitive bias). As such, these measures may be especially useful to use when there is a concern that results may be driven by a specific nuisance variable. Unfortunately, it is currently difficult to determine why these measures show the opposite effects (or, for that matter, why most measures show the dependencies they show). Understanding the nature of these relationships will likely require further progress in developing well-fitting process models of metacognition^{53,54}.

Dear Editor

Thank you for your positive response to the previous version of the manuscript. I have now addressed all remaining comments, as detailed in the point-by-point response below. I have also complied with all formatting and reporting requirements. Thank you for all your guidance and support with this manuscript.

Sincerely,
Doby Rahnev

Reviewer #1

The author has produced a very thorough and thoughtful response to the first set of reviews – thank you for being so responsive my original comments. I am convinced by the additions and edits and am now happy to recommend publication. I identified only a few minor issues to address:

Thank you for this assessment and for catching the issues below.

p.24. “meta-uncertainty increased for easier conditions in all three datasets”. This doesn’t seem to be true in the Shekhar dataset, where the value is lowest is lowest in the condition with highest d-prime.

Thank you for noticing that. This statement was based on the data in Figure 4 (currently Figure 2b) on effect sizes, but it’s true that the means plotted in Figure 3 (currently Figure 2a) showed a slightly different pattern. I examined what caused this discrepancy and discovered that one subject in the Shekhar dataset had a NaN value for meta-uncertainty (due to outlier exclusions, as described in the paper). This NaN value appeared only in the easy condition. So, the bars in Figure 2a for the Shekhar dataset previously showed averages of 20 subjects for two of the conditions, but the average of 19 subjects in the last condition. I have now fixed Figure 2, so that NaN values for a given subject in one condition automatically exclude the subject from contributing to the average for all conditions. The resulting figure is almost identical, but it does now show meta-uncertainty increasing for all three datasets, which is consistent with Figure 2b. I’m pasting the updated Figure 2 below:

Figure 2. Dependence of estimated metacognitive scores on task performance. (a) Estimated metacognitive ability for all 17 measures, as well as d' , criterion, and confidence for different difficulty levels in three datasets (Shekhar, Rouault1, and Rouault2). Traditional measures of metacognition (top row) all showed a strong positive relationship with task performance, whereas all difference measures (third row) show a strong negative relationship. Ratio measures (second row) and the two model-based measures (*meta-noise* and *meta-uncertainty*) performed much better but still showed weak relationships with task performance. Error bars showing SEM are displayed on both the x and y axes. ***, $p < 0.001$; **, $p < 0.01$; *, $p < 0.05$; ns, not significant. (b) Effect sizes for dependence on task performance. Effect size (Cohen's d) is plotted for each metric and dataset. As can be seen in the figure, non-normalized traditional measures (i.e., *meta-d'*, *AUC2*, *Gamma*, *Phi*, and ΔConf) show strong positive relationship with task performance. Corrections with the ratio and difference methods reverse this relationship, with the ratio correction being clearly superior. The model-based metrics *meta-noise* and *meta-uncertainty* perform well too, with *meta-uncertainty* showing particularly low effect sizes.

p.45. “A possible disadvantage of both measures is that they can only take positive values and therefore cannot be used for situations where metacognition may contain more information than the decision itself”. It may be helpful to expand this statement, e.g. by giving an intuition or indication what it could mean for metacognition to “contain more information” than a decision, and under what circumstances this can occur.

I agree that a clarification here would be helpful. I have now clarified that this situation occurs most notably when additional information arrives after the decision has already been made. This additional information can make confidence more informative but cannot affect the perceptual decision (which was already made). Below, I paste the updated sentence, where I also include references to papers discussing this phenomenon.

A possible disadvantage of both measures is that they can only take positive values and therefore cannot be used for situations where metacognition may contain more information than the decision itself, such as in the presence of additional information that arrives after the decision^{53,54}.

p.48. “Finally, researchers should interpret findings of M-Ratio being larger (or smaller) than 1 with caution”. I wasn’t sure which aspects of the presented results are the basis for this recommendation. None of the figures include M-Ratio values above 1. Maybe this paragraph is a legacy from analyses in the previous version and needs to be cut in this revision?

This is the last paragraph on a section where I make recommendations about the use and interpretation of different measures. This section brings together the data from the current paper and data and arguments from the previous literature. In the case of this specific paragraph, I think it’s necessary to address the issue of how one should interpret values of M-Ratio being larger than 1 because this topic is subject to debate in the literature. I’ve edited lightly this paragraph for clarity but have otherwise decided to keep it as it addresses an important issue of controversy, and I would like for the current paper to be as comprehensive as possible.

Reviewer #2

The author has responded constructively to my previous comments and the comments of the other reviewers, which has improved the manuscript.

Thank you for this assessment and for the follow-up comment on the ICC analysis, which I address below.

Only one of my previous comments was not quite addressed: “When testing for reliability in test-retest or split-half data, an absolute agreement intraclass correlation coefficient (ICC; Shrout & Fleiss, 1979) is more appropriate than the Pearson correlation coefficient... The

absolute agreement ICC, as the name suggests, tests $y = x$, how well x predicts y assuming they should be the same.”

An ICC was added, but it was a consistency ICC, not an absolute agreement ICC. The consistency ICC also discounts changes in the mean, so would suggest high test-retest reliability if most subjects showed significantly decreased (or increased) metacognitive sensitivity from test 1 to test 2 but by similar amounts. Normally I would consider test-retest reliability to be high if subjects show the same scores from test 1 to test 2, this is what the absolute agreement ICC tests for. In the package the author used, this model is ‘A-1’. The consistency ICC gives very similar results to the Pearson correlation because it is a very similar test.

Thank you for catching this issue. I was unsure which is the most appropriate ICC analysis to run and selected one based on what I have seen in previous papers. However, I see that this was not the appropriate choice. I have correspondingly now re-done the analyses using the ‘A-1’ model instead of the ‘C-1’. The results nevertheless remained very similar to the results from the Pearson correlation. I have pasted the updated figure below:

Figure 6. Test-retest reliability of metacognitive scores. Test-retest correlations in the Haddara dataset (6 days, 500 trials per day, 70 subjects) show generally low test-retest reliability. Upper panel shows ICC values, whereas lower panel shows Pearson correlation. The test-retest reliability was low-to-moderate for the measures *meta-d*, *AUC2*, *Gamma*, *Phi*, and Δ *Conf* and very low for the remaining measures.

Reviewer #4

I greatly enjoyed reading this article, which makes an important contribution to the literature by comparing several metrics aimed at best quantifying metacognitive performance. It will be a handy resource for the community, which is often at a loss when choosing a particular metric. The author has already done a great deal of work to explain the complex concepts discussed in the article. He has also gone to great lengths to respond to reviewers. The result is excellent, and I have only minor comments to make, which you'll find below.

I thank the reviewer for the positive comments and for the constructive comments below. I address each of them in turn.

General Comments:

1. Although mentioned and discussed satisfactorily in the Discussion section (Limitations), the fact that most datasets and measures are designed for two-choice tasks seems relevant. I believe it should also at least be mentioned when introducing the datasets on page 16.

I agree that mentioning this when introducing the datasets would be helpful. I've now added the following sentence to the Introduction: "All datasets involve 2-choice tasks because most measures of metacognition only apply to 2-choice tasks."

2. Page 9 when introducing the second nuisance variable, you have the sentence: "The logic here is similar to the logic in SDT, where the measure of performance is designed to be mathematically independent of the measure of response bias." In my opinion, it makes the text confusing to have response bias used here before it is introduced, even more so when talking about metacognitive bias for the first time. Consider switching the order of introducing the variables metacognitive bias and response bias.

I agree and have now switched the order in which the nuisance variable of response and metacognitive bias are introduced. I've made light edits to the two corresponding paragraph to accommodate the new order and have also updated the order of these two nuisance variables in Table 2 (which has not become Supplementary Table 2).

3. Figure 7 (Page 36), I find this Figure hard to read due to the x-axis labels. Perhaps it would be best to use the 7 numerical categories introduced in the text on page 35 (1 to 7).

I agree and have now updated the figure (now Figure 4) accordingly:

Figure 4. Dependence of estimated metacognitive scores on response bias. (a) Estimated metacognitive ability for all 17 measures, as well as d' , criterion, and confidence for the seven conditions in the Locke dataset. As expected, condition strongly affected response criterion, c. Despite that, condition did not significantly modulate any of the 17 measures of metacognition. The seven conditions in the graph are arranged based on their average criterion values. Error bars show SEM. ***, $p < 0.001$; ns, not significant. (b) Correlation with absolute response bias. Average correlation between estimated metacognitive ability and absolute response bias (i.e., $|c|$) for all 17 measures. As can be seen from the figure, all relationships are relatively small, but there is still a fair amount of uncertainty around each value. Error bars show SEM.

4. Page 37, Figure 7's caption says "While condition strongly modulated response bias..." This is expected, right? If I understand correctly, this result is like a sanity check to compare response bias in the experimental setup (condition) and response bias as a metric (criterion, c), but please clarify if I am understanding something wrongly. Having said this, I have a concern with the mix between the words response bias (criterion, c) and the response bias associated with condition in the experimental setup. I believe it would be a good idea to be careful with the naming convention and avoid ambiguities, given that it is a dense and

complex text. Would it be better to always call criterion (c), criterion, and just use response bias to talk about the condition in the experimental setup? In any case, I think clarification would help. For example, the caption in Figure 7 would then change to “While response bias strongly modulated criterion, c, (...)”.

The reviewer is correct that the sentence “While condition strongly modulated response bias” was meant to communicate an expected result that serves as a sanity check. I see that this is not clear in the current wording and have therefore edited this part. The updated section reads, “As expected, condition strongly affected response criterion, c. Despite that, condition did not significantly modulate any of the 17 measures of metacognition.”

I also agree that I’ve been using the phrase “response bias” to refer both to the experimental manipulation and the measure we use (criterion, c). I have now fixed that and never use the phrase “response bias” to refer to the experimental setup. Instead, I use the phrases “experimental condition” or “condition”.

5. The author newly incorporated a correlation test (ICC) in addition to the Pearson correlation for test-retest reliability. Figure 10 seems to show the same graph for both measurements, could this be a mistake? Otherwise, if the graphs are almost identical, I find the Pearson correlation figure to be unnecessary and could be removed from the paper.

I see the reviewer’s point. However, previous studies used either ICC (Kopcanova et al., 2023) or Pearson correlation (Guggenmos, 2021), and there are lingering questions about whether the two methods would produce similar or different results (see, for example, the comment from Reviewer 2 above). I have therefore elected to keep both tests in the figure, as I believe that this will help the reader make sense of the results and relate them to the previous literature.

Typos and minor

1. Page 6, Table 1 is either missing a full-stop (period) in the second row.

Fixed.

3. I see an inconsistency in the use of decimal values within the text, for example, In page 44, the across subject correlation is reported as “average $r=.49, .55, (...)$ ”, but the inter-measures correlation as “ $r = 0.60, 0.63, (...)$ ”. As said before, the complexity of the topic requires attention and good understanding, I believe it would be easier to read if this was consistent, personally prefer 0.xx format.

Thank you for noticing this. I've now made sure the entire paper uses the 0.xx format. (By the way, there was no comment #2. I'm just mentioning this, so it's clear that I didn't ignore any comments.)

4. Figure 12m Page 48. There is an extra comma, where it says "The figure lists, the values (...)" it should say "The figure lists the values (...)"

Fixed.

5. I would recommend using the symbol for d' consistently throughout the text. Sometimes it is d' and sometimes it is $d'.$

Thank you for catching that. I've now made sure that d' is consistently used throughout.

Reviewer #5

Thank you for the co-review.